# Local Metric Learning for Off-Policy Evaluation in Contextual Bandits with Continuous Actions

**Haanvid Lee[1], Jongmin Lee[2], Yunseon Choi[1], Wonseok Jeon[†],**
**Byung-Jun Lee[3,4], Yung-Kyun Noh[5,6], Kee-Eung Kim[1]**
[1]KAIST, [2]UC Berkeley, [3]Korea Univ., [4]Gauss Labs Inc., [5]Hanyang Univ., [6]KIAS
haanvid@kaist.ac.kr, jongmin.lee@berkeley.edu, cys9506@kaist.ac.kr
byungjunlee@korea.ac.kr, nohyung@hanyang.ac.kr, kekim@kaist.ac.kr

## Abstract

We consider local kernel metric learning for off-policy evaluation (OPE) of deterministic policies in contextual bandits with continuous action spaces. Our work is motivated by practical scenarios where the target policy needs to be deterministic due to domain requirements, such as prescription of treatment dosage and duration in medicine. Although importance sampling (IS) provides a basic principle for OPE, it is ill-posed for the deterministic target policy with continuous actions. Our main idea is to relax the target policy and pose the problem as kernel-based estimation, where we learn the kernel metric in order to minimize the overall mean squared error (MSE). We present an analytic solution for the optimal metric, based on the analysis of bias and variance. Whereas prior work has been limited to scalar action spaces or kernel bandwidth selection, our work takes a step further being capable of vector action spaces and metric optimization. We show that our estimator is consistent, and significantly reduces the MSE compared to baseline OPE methods through experiments on various domains.

## 1 Introduction

In order to deploy a contextual bandit policy to a real-world environment, such as personalized pricing[1], treatments [2], recommendation [3], and advertisements [4], the performance of the policy should be evaluated prior to the deployment to decide whether the trained policy is suitable for deployment. This is because the interaction of the policy with the environment could be costly and/or dangerous. For example, using an erroneous medical treatment policy to prescribe drugs for patients could result in dire consequences. There then emerges a necessity for an algorithm that can evaluate a policy's performance without having it interact with the environment in an online manner. Such algorithms are called "off-policy evaluation" (OPE) algorithms [5]. OPE algorithms for contextual bandits evaluate a target policy by estimating its expected reward from the data sampled by a behavior policy and without the target policy interacting with the environment.

Previous works on OPE for contextual bandits have mainly focused on environments with finite actions [4, 6–10]. The works can be largely divided into three approaches [6, 9, 11]. The first approach is the direct method (DM) which learns an environment model for policy evaluation. DM is known to have low variance [9]. However, since the environment model is learned with function approximation, the estimator is biased. The second approach is importance sampling (IS) which corrects the data distribution induced by a behavior policy to that of a target policy [12], and uses the corrected distribution to approximate the expected reward of the target policy. IS estimates are unbiased when the behavior and target policies are given. However, IS estimates can have large

---

[†]The research was done while in Mila/McGill University, but the author is currently employed by Qualcomm Technologies Inc.

36th Conference on Neural Information Processing Systems (NeurIPS 2022).

variance when there is a large mismatch between the behavior and target policy distributions [9]. The last approach is doubly robust (DR), which uses DM to reduce the variance of IS while keeping the unbiasedness of an IS estimate [6].

Although there are existing works on IS and DR that can be applied to continuous action spaces [13–16], most of them cannot be easily extended to evaluate *deterministic* contextual bandit policies with continuous actions. This is because IS weights used for both IS and DR estimators are almost surely zero for deterministic target policies in continuous action spaces [2]. However, in practice, such OPE algorithms are needed. For example, treatment prescription policies should not stochastically prescribe drugs to patients.

To meet the needs, there are works for evaluating deterministic contextual bandit policies with continuous actions [2, 5, 17]. These works focus on measuring the similarity between behavior and target actions for assigning IS ratios. However, these works either assume a single-dimensional action space [17] which cannot be straightforwardly extended to multiple action dimensions, or, use Euclidean distances for the similarity measures [2, 5]. In general, similarity measures should be learned locally at a state to weigh differences between behavior and target actions in each action dimension differently. For example, in the case of multi-drug prescription, the synergies and side effects of the prescribed drugs are often very complex. Moreover, the different kinds of drugs are likely to have different degrees of effect from person to person [18, 19].

To this end, we propose local kernel metric learning for IS (KMIS) OPE estimation of deterministic contextual bandit policies with multidimensional continuous action spaces. Our proposed method learns a Mahalanobis distance metric [20–23] locally at each state that lengthens or shortens the distance between behavior and target actions to reduce the MSE. The metric-applied kernels measure the similarities between actions according to the Mahalanobis distances induced by the metric. In our work, we first analytically show that the leading-order bias [2] of a kernel-based IS estimator becomes a dominant factor in the leading-order MSE [2] as the action dimension increases given the optimal bandwidth [2] that minimizes the leading-order MSE without a metric. Then we derive the kernel metric that minimizes the upper bound of the leading-order bias, which is bandwidth-agnostic. Our analysis shows that the convergence speed of a kernel-based IS OPE estimator can be improved with the application of the KMIS metrics. In the experiments, we demonstrate that MSEs of kernel-based IS estimators are significantly reduced when combined with the proposed kernel metric learning. We report empirical results in various synthetic domains as well as a real-world dataset.

## 2    Related Work

The works on OPE of deterministic contextual bandits with continuous actions can be largely divided into importance sampling (IS) [2, 5] and doubly robust estimators (DR) [17, 24]. Both methods eliminate the problem of almost surely having zero IS estimates when given a deterministic target policy and a stochastic behavior policy in a continuous action space in two ways. Most of the works relax the deterministic target policy, which can be seen as a Dirac delta function [2], to a kernel [2, 5, 24], and the other work discretizes the action space [17].

Among these, kernel-based IS methods use a kernel to measure the similarities between the target and behavior actions and focus on selecting the bandwidth of the kernel [2, 5, 24]. Su et al. [5] proposed the bandwidth selection algorithm that uses the Lepski's principle for bandwidth selection in the study of nonparametric statistics [25]. Kallus and Zhou [2] derived the leading-order MSE of a kernel-based IS OPE estimation and chose the optimal bandwidth that minimizes the leading-order MSE for the OPE estimation. One of the limitations of the existing kernel-based IS methods [2, 5, 24] is that these methods use Euclidean distances for measuring the similarities between behavior and target actions. The Euclidean distance is inadequate for measuring the similarities as assigning a high similarity measure to the actions having similar rewards will induce less bias (bias definition in Section 3.2). The other limitation is that these methods use one global bandwidth for the whole action space [17]. Since the second-order derivative of the expected reward w.r.t. an action is related to the leading-order bias derived by Kallus and Zhou [2], the optimal bandwidth that balances the bias-variance trade-off may vary depending on actions.

As for discretization methods, Cai et al. [17] proposed deep jump learning (DJL) [17] that avoids the limitation of kernel-based methods due to using one global bandwidth by adaptively discretizing one-dimensional continuous action spaces. The action space is discretized to have similar expected

rewards for each discretized action interval given a state. By using the discretized intervals for DR estimation, DJL can estimate the policy value more accurately in comparison to kernel-based methods when the action space has second-order derivatives of the expected rewards which change significantly across the action space. In such cases, kernel-based methods use the same bandwidth for all actions even though the optimal bandwidth varies depending on actions. On the other hand, DJL can discretize the action space into smaller intervals for the parts of the action space where the second-order derivative is high, and larger intervals when it is low. However, their work focuses on domains with a single action dimension and cannot be easily extended to environments with multidimensional action spaces.

To tackle the limitation of kernel-based methods caused by using Euclidean distances, kernel metric learning can be applied to shrink or extend distances in the directions that reduce MSE. Metric learning has been used for nearest neighbor classification [22, 26–29], and kernel regression [22, 23]. Among them, our work was inspired by the work of Noh et al. that learns a metric for reducing the bias and MSE of kernel regression [23]. In their work, they made the assumption on the generative model of the data where the input and output are jointly Gaussian. Under this assumption, they derived the Mahalanobis metric that reduces the bias and MSE of Nadaraya-Watson kernel regression. We learn our metric in a similar fashion in the context of OPE in contextual bandits except that we do not have the generative assumption on the data as the assumption is unnatural in our problem setting.

## 3 Preliminaries

### 3.1 Problem Setting

In this work, we focus on OPE of a deterministic target policy in an environment with multidimensional continuous action space $\mathcal{A} \subset \mathbb{R}^{D_A}$, state space $\mathcal{S} \subset \mathbb{R}^{D_S}$, reward $r \in \mathbb{R}$ sampled from the conditional distribution of the reward $p(r \mid \mathbf{s}, \mathbf{a})$, state distribution $p(\mathbf{s})$. The *deterministic* target policy $\pi$ can be regarded as having a Dirac delta distribution $\pi(\mathbf{a} \mid \mathbf{s}) = \delta(\pi(\mathbf{s}) - \mathbf{a})$, where the probability density function (PDF) value is zero everywhere except at the selected target action given a state $\pi(\mathbf{s})$. The offline dataset $D = \{(\mathbf{s}_i, \mathbf{a}_i, r_i)\}_{i=1}^{N}$ used for the evaluation of the deterministic target policy $\pi$ is sampled from the environment using a known stochastic behavior policy $\pi_b : \mathcal{S} \to \Delta(\mathcal{A})$. We assume that the support of the behavior policy $\pi_b$ contains the actions selected by the target policy $\pi(\mathbf{s})$. The goal of OPE is to evaluate the target policy value $\rho^{\pi} = \mathbb{E}_{\mathbf{s} \sim p(\mathbf{s}), \mathbf{a} \sim \pi(\mathbf{a} \mid \mathbf{s}), r \sim p(r \mid \mathbf{s}, \mathbf{a})}[r]$ using $D$ and without $\pi$ interacting with the environment.

### 3.2 Bandwidth Selection for the Isotropic Kernel-Based IS Estimator

One of the methods to evaluate the target policy value by using the offline data $D$ sampled with $\pi_b$ is to perform IS estimation. The IS ratios correct the action sampling distribution for the expectation from $\pi_b$ to $\pi$. However, since the density of a deterministic target policy $\pi$ is a Dirac delta function, its PDF value at the behavior action sampled from $\pi_b$ is almost surely zero. Existing works deal with the problem by relaxing the target policy $\pi$ to an isotropic kernel $K$ with a bandwidth $h$ and computing the IS estimate of the policy value $\hat{\rho}^K$ as in Eq. (1) [2, 5, 24].

$$\rho^{\pi} = \mathbb{E}_{\mathbf{s} \sim p(\mathbf{s}), \mathbf{a} \sim \pi_b(\mathbf{a} \mid \mathbf{s}), r \sim p(r \mid \mathbf{s}, \mathbf{a})} \left[ \frac{\pi(\mathbf{a} \mid \mathbf{s})}{\pi_b(\mathbf{a} \mid \mathbf{s})} r \right]$$

$$\approx \frac{1}{N h^{D_A}} \sum_{i=1}^{N} K \left( \frac{\mathbf{a}_i - \pi(\mathbf{s}_i)}{h} \right) \frac{r_i}{\pi_b(\mathbf{a}_i \mid \mathbf{s}_i)}. \tag{1}$$

As the Dirac delta function can be regarded as a kernel having its bandwidth $h$ approaching zero, the relaxation of the Dirac delta function to a kernel can be seen as increasing its $h$. By the relaxation, the bias of the kernel-based IS estimation $\text{Bias}\left[\hat{\rho}^K\right] := \mathbb{E}_{\mathbf{s} \sim p(\mathbf{s}), \mathbf{a} \sim \pi_b(\mathbf{a} \mid \mathbf{s}), r \sim p(r \mid \mathbf{s}, \mathbf{a})}\left[\hat{\rho}^K - \rho^{\pi}\right]$ increases while its variance is reduced. As the bias and the variance compose the MSE of the estimate, the bandwidth that best balances between them and reduce the MSE should be selected. Kallus and Zhou [2] derived the leading-order MSE (LOMSE) in terms of bandwidth $h$, sample size $N$, and action dimension $D_A$ for selecting a bandwidth (Eq. (2)) assuming that $h \to 0$ and $\frac{1}{N h^{D_A}} \to 0$ as $N \to \infty$ (derivation in Appendix A.1). They also derived the optimal bandwidth $h^*$ that minimizes the LOMSE (derivation in Appendix A.2).

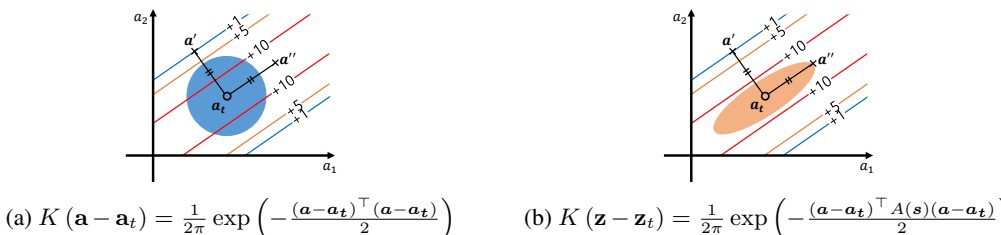

(a) $K\left(\mathbf{a} - \mathbf{a}_t\right) = \frac{1}{2\pi} \exp\left(-\frac{(\mathbf{a}-\mathbf{a}_t)^\top (\mathbf{a}-\mathbf{a}_t)}{2}\right)$  (b) $K\left(\mathbf{z} - \mathbf{z}_t\right) = \frac{1}{2\pi} \exp\left(-\frac{(\mathbf{a}-\mathbf{a}_t)^\top A(\mathbf{s})(\mathbf{a}-\mathbf{a}_t)}{2}\right)$

Figure 1: Illustration of bias reduction in a kernel-based IS estimate by the metric $A(\mathbf{s})$ locally learned at a given state $\mathbf{s}$. The contour line is drawn for the reward over the action space given $\mathbf{s}$. Although behavior actions $\mathbf{a}'$ and $\mathbf{a}''$ are away from target action $\mathbf{a}_t$ ($= \pi(\mathbf{s})$) by an equal Euclidean distance, their corresponding rewards are different. When an (a) isotropic Gaussian kernel is used, bias can come from $\mathbf{a}'$ since the similarity measures of $\mathbf{a}'$, and $\mathbf{a}''$ from $\mathbf{a}_t$ are the same even though their rewards are different. However, when the (b) metric is applied, bias is reduced as the similarity measure is higher on $\mathbf{a}''$ that has a similar reward to $\mathbf{a}_t$ compared to that of $\mathbf{a}'$.

$$\text{LOMSE}(h, N, D_A) = \underbrace{h^4 C_b}_{\text{(leading-order bias)}^2} + \underbrace{\frac{C_v}{N h^{D_A}}}_{\text{(leading-order variance)}}, \tag{2}$$

$$h^* = \arg\min_h \text{LOMSE}(h, N, D_A) = \left(\frac{D_A C_v}{4 N C_b}\right)^{\frac{1}{D_A + 4}}, \tag{3}$$

$$C_b := \frac{1}{4}\mathbb{E}_{\mathbf{s} \sim p(\mathbf{s})}\left[\nabla^2_{\mathbf{a}} r(\mathbf{s}, \mathbf{a})\,|_{\mathbf{a}=\pi(\mathbf{s})}\right]^2, \quad C_v := R(K)\mathbb{E}_{\mathbf{s} \sim p(\mathbf{s})}\left[\frac{\mathbb{E}[r^2 | \mathbf{s}, \mathbf{a} = \pi(\mathbf{s})]}{\pi_b(\mathbf{a} = \pi(\mathbf{s}) | \mathbf{s})}\right],$$

where the first term in Eq. (2) is the squared leading-order bias and the second term is the leading-order variance, $C_b$ and $C_v$ are constants related to the leading-order bias and variance, respectively, the expected reward is $r(\mathbf{s}, \mathbf{a}) := \mathbb{E}[r | \mathbf{s}, \mathbf{a}]$, $\nabla^2_{\mathbf{a}}$ denotes the Laplacian operator w.r.t. action $\mathbf{a}$, the roughness of the kernel is $R(K) := \int K(\mathbf{u})^2 d\mathbf{u}$. The kernel used for the derivation satisfies $\int K(\mathbf{u}) d\mathbf{u} = 1$ and $K(\mathbf{u}) = K(-\mathbf{u})$ for all $\mathbf{u}$. For simplicity, we assumed a Gaussian kernel and used the property $\int \mathbf{u}\mathbf{u}^\top K(\mathbf{u}) d\mathbf{u} = I$.

In Section 4, we use the leading-order bias and variance [2] for the derivation of our proposed metric. We also use the optimal bandwidth [2] for analyzing the properties of kernel-based IS estimator with and without a metric.

### 3.3 Mahalanobis Distance Metric

Relaxing the Dirac delta target policy $\pi$ to a kernel $K$ prevents the kernel-based IS estimator from estimating zero almost surely. By using an isotropic kernel for the relaxation, the difference between the target and behavior actions in all directions are treated equally for measuring the similarities between the actions with a kernel in Eq. (1). However, to produce a more accurate OPE estimation, the difference between the two actions in some directions should be ignored relative to the others for measuring the similarity between the actions. For this, the Mahalanobis distance can be used. We define the Mahalanobis distance between two $D_A$-dimensional vectors $\mathbf{a}_i \in \mathbb{R}^{D_A}$ and $\mathbf{a}_j \in \mathbb{R}^{D_A}$ with the metric $A \in \mathbb{R}^{D_A \times D_A}$ as in Eq. (4). Applying the Mahalanobis distance metric $A\,(= LL^\top)$ to a kernel function can be seen as linearly transforming the kernel inputs with the transformation matrix $L$ $(L^\top \mathbf{a} = \mathbf{z})$.

$$\|\mathbf{a}_i - \mathbf{a}_j\|_A := \sqrt{(\mathbf{a}_i - \mathbf{a}_j)^\top A(\mathbf{a}_i - \mathbf{a}_j)} = \|\mathbf{z}_i - \mathbf{z}_j\|, \quad (A \succ 0,\ A^\top = A,\ |A| = 1). \tag{4}$$

Figure 1 illustrates a case where the Mahalanobis metric is locally learned at a given state for reducing the bias of a kernel-based IS estimate. The bias of the estimate is reduced by altering the shape of the kernel to produce a higher similarity measure on a behavior action that has a similar reward to the target action.

## 4 Local Metric Learning for Kernel-Based IS

### 4.1 Local Metric Learning via LOMSE Reduction

To reduce the LOMSE (Eq. (2)) of the kernel-based IS estimate by learning a metric, we first analyze the LOMSE when the optimal bandwidth (Eq. (3)) is applied to the kernel. In Figure 1 we illustrate

how the bias of an IS estimation is induced due to kernel relaxation. The bias can worsen in high dimensional spaces. To analytically show this, we adapt Proposition 4 in the work of Noh et al. [23] and show that the $C_b$ contained in the squared leading-order bias (Eq. (2)) becomes a dominant term in the LOMSE of kernel-based IS OPE given an optimal bandwidth and as the action dimension increases.

**Proposition 1.** *(Adapted from Noh et al. [23]) For a high dimensional action space $D_A \gg 4$, and given optimal bandwidth $h^*$ (Eq. (3)), the squared leading-order bias dominates over the leading-order variance in LOMSE. Furthermore,* LOMSE($h^*, N, D_A$) *can be approximated by $C_b$ in Eq. (2).*

$$\text{LOMSE}(h^*, N, D_A) = N^{-\frac{4}{D_A+4}} \left( \left(\frac{D_A}{4}\right)^{\frac{4}{D_A+4}} + \left(\frac{4}{D_A}\right)^{\frac{D_A}{D_A+4}} \right) C_b^{\frac{D_A}{D_A+4}} C_v^{\frac{4}{D_A+4}} \approx C_b. \quad (5)$$

The proof is made by plugging in Eq. (3) to Eq. (2) and taking the limit $D_A \to \infty$. (See Appendix B.1.) Proposition 1 implies that in an environment with high dimensional action space, the MSE can be significantly reduced by the reduction of $C_b$ contained in the squared leading-order bias (Eq. (2)). Therefore, we aim to reduce $C_b$, or, reduce the leading bias in a bandwidth-agnostic manner with a Mahalanobis distance metric that shortens the distance between the target and behavior actions in the direction where their corresponding rewards are similar.

As mentioned in Section 3.3, applying a state-dependent metric ($A(\mathbf{s}) = L(\mathbf{s})L(\mathbf{s})^\top$) to a kernel is equivalent to linearly transforming input vectors of the kernel. Therefore, $C_b$ with a metric $A : \mathcal{S} \to \mathbb{R}^{D_A \times D_A}$ (i.e. $C_{b,A}$) should be analyzed with the linearly transformed actions $L(\mathbf{s})^\top \mathbf{a}$ (derivation in Appendix A.3), and we aim to find an optimal $A$ that minimizes it:

$$\min_{\substack{A: A(\mathbf{s}) \succ 0, \\ A(\mathbf{s}) = A(\mathbf{s})^\top, |A(\mathbf{s})| = 1 \, \forall \mathbf{s}}} C_{b,A} = \frac{1}{4} \mathbb{E}_{\mathbf{s} \sim p(\mathbf{s})} \left[ \text{tr} \left( A(\mathbf{s})^{-1} \, \mathbf{H_a} \, r(\mathbf{s}, \mathbf{a}) \big|_{\mathbf{a} = \pi(\mathbf{s})} \right) \right]^2, \quad (6)$$

where $\mathbf{H}$ is the Hessian operator. In the derivation, we assume a Gaussian kernel and use the property $\int \mathbf{u}\mathbf{u}^\top K(\mathbf{u}) d\mathbf{u} = I$. Still, optimizing the function $A$ in Eq. (6) itself is challenging since it requires considering the overall effect of each metric matrix $A(\mathbf{s})$ on the objective function. Therefore, we instead consider minimizing the following upper bound, which allows us to compute the closed-form metric matrix for each state in a nonparametric way:

$$\min_{\substack{A: A(\mathbf{s}) \succ 0, \\ A(\mathbf{s}) = A(\mathbf{s})^\top, |A(\mathbf{s})| = 1 \, \forall \mathbf{s}}} U_{b,A} = \frac{1}{4} \mathbb{E}_{\mathbf{s} \sim p(\mathbf{s})} \left[ \text{tr} \left( A(\mathbf{s})^{-1} \, \mathbf{H_a} \, r(\mathbf{s}, \mathbf{a}) \big|_{\mathbf{a} = \pi(\mathbf{s})} \right)^2 \right]. \quad (7)$$

In the following Theorem 1, we introduce the optimal Mahalanobis metric matrix $A^*(\mathbf{s})$ that minimizes $U_{b,A}$.

**Theorem 1.** *(Adapted from Noh et al. [22]) Assume that the $p(r|\mathbf{s}, \mathbf{a})$ is twice differentiable w.r.t. an action $\mathbf{a}$. Let $\Lambda_+(\mathbf{s})$ and $\Lambda_-(\mathbf{s})$ be diagonal matrices of positive and negative eigenvalues of the Hessian $\mathbf{H_a} \mathbb{E}[r|\mathbf{s}, \mathbf{a}]|_{\mathbf{a}=\pi(\mathbf{s})}$, $U_+(\mathbf{s})$ and $U_-(\mathbf{s})$ be matrices of eigenvectors corresponding to $\Lambda_+(\mathbf{s})$ and $\Lambda_-(\mathbf{s})$ respectively, and $d_+(\mathbf{s})$ and $d_-(\mathbf{s})$ be the numbers of positive and negative eigenvalues of the Hessian. Then the metric $A^*(\mathbf{s})$ that minimizes $U_{b,A}$ is:*

$$A^*(\mathbf{s}) = \alpha(\mathbf{s}) \left[ U_+(\mathbf{s}) U_-(\mathbf{s}) \right] \underbrace{\begin{pmatrix} d_+(\mathbf{s})\Lambda_+(\mathbf{s}) & 0 \\ 0 & -d_-(\mathbf{s})\Lambda_-(\mathbf{s}) \end{pmatrix}}_{=:M(\mathbf{s})} \left[ U_+(\mathbf{s}) U_-(\mathbf{s}) \right]^\top, \quad (8)$$

*where $\alpha(\mathbf{s}) := |M(\mathbf{s})|^{-1/(d_+(\mathbf{s})+d_-(\mathbf{s}))}$.*

The proof involves solving a Lagrangian equation to minimize the square of the trace term in Eq. (7) with constraints on $A^*(\mathbf{s})$. (See Appendix B.2.) In the special case where the Hessian matrix contains both positive and negative eigenvalues for all states, the metric reduces the $C_{b,A}$ to zero. (See Appendix B.2.) $A^*(\mathbf{s})$ is locally computed at a state $\mathbf{s}$ and the corresponding target action $\pi(\mathbf{s})$ as the optimal metric is derived from the Hessian at that point $\mathbf{H_a} \, r(\mathbf{s}, \mathbf{a})|_{\mathbf{a}=\pi(\mathbf{s})}$. When the optimal metric is applied to the kernel, it measures the similarity between a behavior and target action using the Mahalanobis distance instead of the Euclidean distance, as shown in Figure 1 and in Eq. (9):

$$\|\mathbf{a} - \pi(\mathbf{s})\|_{A^*(\mathbf{s})}^2 = \alpha(\mathbf{s}) \sum_{i=1}^{d_+} \left\{ \sqrt{d_+(\mathbf{s})\lambda_{+,i}(\mathbf{s})} \mathbf{u}_{+,i}(\mathbf{s})^\top (\mathbf{a} - \pi(\mathbf{s})) \right\}^2 \quad (9)$$

$$+ \alpha(\mathbf{s}) \sum_{i=1}^{d_-} \left\{ \sqrt{-d_-(\mathbf{s})\lambda_{-,i}(\mathbf{s})} \mathbf{u}_{-,i}(\mathbf{s})^\top (\mathbf{a} - \pi(\mathbf{s})) \right\}^2,$$

where $\lambda_{+,i}(\mathbf{s})$ and $\lambda_{-,i}(\mathbf{s})$ are positive and negative eigenvalues of the Hessian $\mathbf{H_a}\, r(\mathbf{s}, \mathbf{a})|_{\mathbf{a}=\pi(\mathbf{s})}$, and their corresponding eigenvectors are denoted as $\mathbf{u}_{+,i}(\mathbf{s})$ and $\mathbf{u}_{-,i}(\mathbf{s})$ respectively. The Mahalanobis distance is formed to weigh the difference between the behavior action $\mathbf{a}$ and the target action $\pi(\mathbf{s})$ w.r.t. the axes formed by the eigenvectors. $A^*(\mathbf{s})$ increases the distance between the actions in the direction of eigenvectors whose corresponding eigenvalues are large, and vice versa.

Now we analyze how the optimal metric affects the convergence rate of a kernel-based IS estimator w.r.t. data size $N$ and the action dimension $D_A$ similar to Theorem 3 in the work of Kallus and Zhou [2].

**Theorem 2.** *(Adapted from Kallus and Zhou [2]) Kernel-based IS estimator with the optimal metric $A^*(\mathbf{s})$ from Eq. (8) and the optimal bandwidth $h^*$ in Eq. (3) is a consistent estimator in which convergence rate is faster than or equal to that of the isotropic kernel-based IS estimator with $h^*$. When the Hessian $\mathbf{H_a}\, r(\mathbf{s}, \mathbf{a})|_{\mathbf{a}=\pi(\mathbf{s})}$ has both positive and negative eigenvalues for all states, $A^*(\mathbf{s})$ applied estimator converges to the true policy value faster than the one without the metric by the rate of $\mathcal{O}(D_A^{-\frac{1}{2}})$ as the action dimension increases.*

The proof is made by analyzing the complexity of the LOMSE with and without the proposed metric w.r.t. both sample size $N$ and action dimension $D_A$. (See Appendix B.3.)

## 4.2 Practical Algorithm

To avoid the degenerate case of having zero for all eigenvalues of the Hessian $\mathbf{H_a}\, r(\mathbf{s}, \mathbf{a})|_{\mathbf{a}=\pi(\mathbf{s})}$, we add in the regularizer $\gamma(\mathbf{s})$, and also use $\beta(\mathbf{s})$ to make $|\hat{A}(\mathbf{s})| = 1$ as in Eq. (10) so that the metric applied kernel can fall back to an isotropic kernel in the degenerate case. (See Appendix C.1 for more details.) For the simplicity of the algorithm, we did not discard the components along the eigenvectors with zero eigenvalues ($U_0(\mathbf{s})$) but let the metric have longer relative bandwidths in those directions compared to the others by the regularizers. The KMIS metric $\hat{A}(\mathbf{s})$ with the regularizers is,

$$\hat{A}(\mathbf{s}) = \beta(\mathbf{s})\,[U_+(\mathbf{s})U_-(\mathbf{s})U_0(\mathbf{s})] \begin{pmatrix} d_+(\mathbf{s})\Lambda_+(\mathbf{s}) & 0 & 0 \\ 0 & -d_-(\mathbf{s})\Lambda_-(\mathbf{s}) & 0 \\ 0 & 0 & \mathbf{0} \end{pmatrix} [U_+(\mathbf{s})U_-(\mathbf{s})U_0(\mathbf{s})]^\top + \gamma(\mathbf{s})I. \tag{10}$$

To apply the KMIS metric $\hat{A}(\mathbf{s})$ on a kernel-based IS estimation, we first fit a neural network reward regressor with the dataset $D$ for the estimation of the Hessian matrix $\mathbf{H_a}\, r(\mathbf{s}, \mathbf{a})|_{\mathbf{a}=\pi(\mathbf{s})}$. The reward regressor may have some error in the estimation of the Hessian and may have an adverse effect on our algorithm. (Effect analyzed in Appendix F.1.) Using the estimated Hessian, we linearly transform the kernel inputs with the transformation matrix $\hat{L}(\mathbf{s})$ ($\hat{A}(\mathbf{s}) = \hat{L}(\mathbf{s})\hat{L}(\mathbf{s})^\top$) in Eq. (11). Then given a kernel, and a bandwidth from the previous works on kernel-based IS that selects bandwidths [2][5], IS estimation can be made with the offline data $D$ as in Algorithm 1.

$$\hat{L}(\mathbf{s}) = [U_+(\mathbf{s})U_-(\mathbf{s})U_0(\mathbf{s})] \left[ \begin{pmatrix} \beta(\mathbf{s})d_+(\mathbf{s})\Lambda_+(\mathbf{s}) & 0 & 0 \\ 0 & -\beta(\mathbf{s})d_-(\mathbf{s})\Lambda_-(\mathbf{s}) & 0 \\ 0 & 0 & \mathbf{0} \end{pmatrix} + \gamma(\mathbf{s})I \right]^{\frac{1}{2}}. \tag{11}$$

---

**Algorithm 1** KMIS for Kernel-Based IS Estimation

---

**Input:** Offline data $D = \{\mathbf{s}_i, \mathbf{a}_i, r_i\}_{i=1}^N$, behavior policy $\pi_b$, deterministic target policy $\pi$, reward function parameters $\phi$, bandwidth $h$, kernel $K$.
**Output:** Estimate of the target policy value $\hat{\rho}^K$
1: Fit the neural network regressor $r_\phi(\mathbf{s}, \mathbf{a})$ with $D$
2: Compute $\mathbf{H_a}\, r_\phi(\mathbf{s}, \mathbf{a})|_{\mathbf{a}=\pi(\mathbf{s})}$
3: Compute $\hat{L}(\mathbf{s})$ in Eq. (11) from $\mathbf{H_a}\, r_\phi(\mathbf{s}, \mathbf{a})|_{\mathbf{a}=\pi(\mathbf{s})}$
4: Transform the kernel inputs: $\mathbf{z}_i = \hat{L}(\mathbf{s}_i)^\top (\mathbf{a}_i - \pi(\mathbf{s}_i))$, for all $i$
5: Compute the estimate: $\frac{1}{Nh^{D_A}} \sum_{i=1}^N K\left(\frac{\mathbf{z}_i}{h}\right) \frac{r_i}{\pi_b(\mathbf{a}_i|\mathbf{s}_i)}$

---

# 5 Experiments

In this section, we show that applying our KMIS metric reduces the MSEs of the kernel-based IS estimators with bandwidths selected from existing works [2, 5] on synthetic domains and Warfarin dataset [30]. For baselines, we use existing works on kernel-based IS [2, 5] and a direct method (DM). For synthetic domains, we also include a simple discretized OPE estimator [2] as a baseline. For the baselines of existing works of kernel-based IS, we use the work of Kallus and Zhou [2] and SLOPE [5] which are bandwidths selection algorithms for kernel-based IS. For the direct method (DM) we use a neural network reward regressor with a Gaussian output layer. The discretized OPE estimator [2] evenly discretizes the action space by 10 for each action dimension (resulting in 100 bins in 2-dimensional action spaces) for an IS estimation. The baselines are compared to our proposed KMIS metric applied kernel-based IS estimator with bandwidths selected from Kallus and Zhou's estimator or SLOPE. DM's reward regressor is used for estimating the Hessian $\mathbf{H_a}\, r(\mathbf{s}, \mathbf{a})|_{\mathbf{a}=\pi(\mathbf{s})}$ (Eq. (8)) required for KMIS metric computation and the optimal bandwidth selection of Kallus and Zhou's estimator (compute $C_v$ and $C_b$ in Eq. (3) with DM). We use DM for bandwidth selection of Kallus and Zhou's estimator instead of using kernel density estimation (KDE), which is used in the work of Kallus and Zhou [2], since using KDE to estimate the bandwidth is computationally expensive [17]. For all estimators, we use self-normalization as in the work of Kallus and Zhou [2] as it is known to reduce estimation variance significantly with the addition of a small bias and result in reduced MSE. We do not correct the boundary bias as opposed to the works of Kallus and Zhou [2], Su et al. [5] since we regard the bias induced by the boundaries are negligible compared to the bias reduced by our metric. For more details of the experiments, see Appendix E.

## 5.1 Synthetic Data

In the experiment with the synthetic data, we show that our KMIS metric can be applied to the offline data sampled from environments with various reward functions. With such offline data, we first show the MSEs of existing kernel-based IS estimators can be reduced by our KMIS metric. Furthermore, we empirically validate Proposition 1. Lastly, we visualize the learned KMIS metrics to see if the metrics are learned as we intended.

We prepare three synthetic domains which are quadratic reward domain, absolute error domain, and multi-modal reward domain. All synthetic domains use actions and states in $\mathbb{R}^2$. In the quadratic reward domain, the rewards of the offline data are sampled from a normal distribution, where the mean is from a quadratic function w.r.t. states and actions (for details, see Appendix E). Each dimension of states are independently and uniformly sampled in the range of $[-1, 1]$, actions are sampled from $\pi_b(\mathbf{a}\,|\,\mathbf{s}) = N(\mathbf{s} + 0.2I, 0.5^2 I)$, target policy is $\pi(\mathbf{s}) = \mathbf{s}$. If the metric is learned successfully from the data, the metric should form an ellipsoidal shape and should be the same for any state and action since the Hessian $\mathbf{H_a}\, r(\mathbf{s}, \mathbf{a})$ of the quadratic reward function is a constant.

In the absolute error domain, the offline data is sampled from the environment with deterministic rewards given states and actions $r(\mathbf{s}, \mathbf{a}) = -|0.5s_1 - a_1|$, where $s_1$ and $a_1$ are the first dimensions of the state and action vectors. Each dimension of both states and actions are independently and uniformly sampled in the range of $[-1, 1]$, and the target policy is $\pi(\mathbf{s}) = 0.5\mathbf{s}$. Since the absolute value function has points in its domain where it is not twice differentiable, and the Hessian $\mathbf{H_a}\, r(\mathbf{s}, \mathbf{a})$ is zero at the twice differentiable points, it is reasonable to think that the metric learning would fail in such cases. However, since we are dealing with finite samples and using a neural network reward regressor, we conjecture that some form of concave-shaped reward estimate can be learned by the reward regressor and make meaningful metric learning possible. If the conjecture is right and the metric learning is successful, the metric will be learned to have a relatively larger bandwidth in the direction of the dummy action dimension ($a_2$) that is unrelated to the reward.

Lastly, for the multi-modal reward domain, the multi-modal reward function is made with exponential functions and max operators similar to the multi-modal reward function introduced in the work of Haarnoja et al. [31]. Rewards are deterministic given states and actions. (For the details see Appendix E.) Each dimension of states and actions are independently and uniformly sampled from the range of $[-1, 1]$, and the target policy is $\pi(\mathbf{s}) = \mathbf{s} + \left[\begin{smallmatrix} 0.5 \\ 0 \end{smallmatrix}\right]$. Due to the max operators, there are points in the reward function domain where it is not twice differentiable. For the twice differentiable points, the reward function is designed to have varying Hessians $\mathbf{H_a}\, r(\mathbf{s}, \mathbf{a})$, thus, varying metrics should be learned for varying actions unlike the other synthetic domains. Also, since the Taylor

expansion terms of the reward function w.r.t. the target action have higher order terms beyond 2nd order, the bias of the kernel-based IS estimation without the metric will contain the terms ignored in the derivation of the squared leading-order bias in Eq. (2) (the derivation in Appendix A.1) which upper bound is minimized by the KMIS metric.

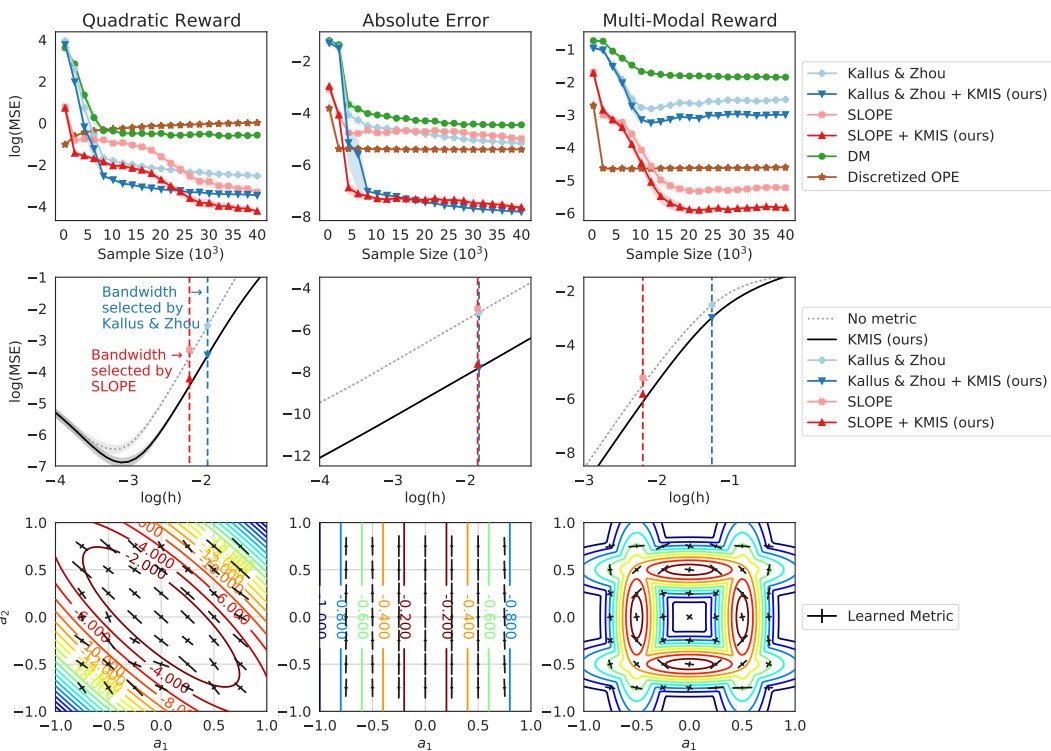

Figure 2: Experimental results on the synthetic domains. Columns starting from the left are results from the quadratic reward domain, absolute error domain, and multi-modal reward domain. The first row shows the performance of estimators as the data size is increased. The second row shows the amount of MSEs reduced by the KMIS metrics given various bandwidths with 40k data points. The second row also marks the results of the first row when the data size used in the estimation is 40k. For both first and second row figures, means and standard errors of squared errors from 100 trials are drawn. The last row visualizes the learned metrics from a trial along with a reward landscape.

The experimental results on the synthetic domains are shown in Figure 2. The first row of Figure 2 shows that for most cases, the KMIS metric reduces the MSEs of the kernel-based IS estimators. Also, the KMIS metric applied kernel-based IS estimators outperform DM, which is used by the KMIS algorithm to estimate the Hessians of a reward function for the metric learning. For the multi-modal reward domain, even though the bias contains terms ignored in the leading-order bias, the metrics that minimize the upper bound of squared leading-order bias also reduce MSE. The discretized OPE estimator performs worst in the quadratic reward domain for the sample size above 10k, but it performs better than some other estimators in the other domains. As it is unclear how to discretize the multidimensional action space [2], even though the same discretization rule is used for all synthetic domains, its performance varies from domain to domain.

The second row of Figure 2 shows that for most of the given bandwidths, the MSEs of the kernel-based IS estimations are reduced by the KMIS metrics. The gray dotted line denotes the MSEs of kernel-based IS estimators with given bandwidths and without a metric. The black line denotes the KMIS metric applied kernel-based IS estimators with given bandwidths. The dotted vertical lines present the average of the selected bandwidths by SLOPE (red) or Kallus and Zhou's estimator (blue), and the markers show the average MSEs of the estimators at the average bandwidths when 40k data points are used for estimation. The average MSEs of KMIS metric applied estimators with the same 40k data points are also marked. Since the selected bandwidths can vary for each run, the markers may not exactly lie on the black lines denoting the results of the fixed bandwidths.

The third row of Figure 2 shows learned KMIS metrics from a trial and the reward landscape when $\mathbf{s} = \begin{bmatrix} 0 \\ 0 \end{bmatrix}$ for each synthetic domain. The learned metrics are drawn with black crosses. The metrics are not only drawn for a target action but also for other actions. For the metrics learned in the quadratic reward domain, the learned metrics are ellipsoidal and similar for most actions. Therefore, we can see that our metric is learned as intended with the Hessian $\mathbf{H_a}\, r(\mathbf{s}, \mathbf{a})$ provided by the reward regressor (DM). For the absolute error domain, the metrics are learned to ignore the dummy action dimension $a_2$ by having longer relative bandwidth in the direction of the action dimension while having shorter relative bandwidth in the direction of $a_1$ which is used for computing the reward. This result empirically verifies our conjecture made earlier, which suggests that our metric learning algorithm can be generally applied to the dataset where the true reward function is not twice differentiable at some actions, and the true Hessian is zero at the twice differentiable actions. The metrics learned on the multi-modal reward made with exponents verifies that our metric indeed learns varying metrics according to actions.

To empirically validate Proposition 1, the empirical squared bias, variance, and MSE ($\mathrm{MSE}[\widehat{\rho}^K] = \mathrm{Bias}[\widehat{\rho}^K]^2 + Var[\widehat{\rho}^K]$) of the Kallus and Zhou's estimator and the KMIS metric applied version of the estimator is observed on the modified absolute error domain with various number of dummy action dimensions. The modified domain can have additional dummy action dimensions where the value of each dummy dimension is independently and uniformly sampled in the range of $[-1, 1]$. The result in Figure 3 shows that the empirical squared bias of Kallus and Zhou's estimator is indeed the dominant term in the MSE in high action dimensions. As the action dimension is increased, both algorithms suffer from high bias. But the KMIS metric reduces the bias and shows lower MSE than the one without the metric. However, the KMIS metric applied estimator shows higher variance than the Kallus and Zhou's estimator. This increase in the variance due to the metric may come from the ignored components of the variance, or, the Hessian estimation error.

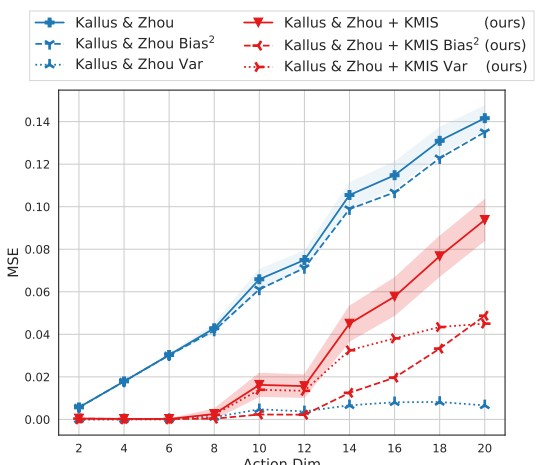

Figure 3: OPE performance on the modified absolute error domain with various number of action dimensions. Means and standard errors of squared errors were obtained from 100 trials with 40k samples. Empirical squared bias and variance of the estimates are also drawn.

## 5.2 Warfarin Data

To test our algorithm on a more realistic dataset, we test our algorithm on the Warfarin dataset [30]. Warfarin is a treatment that is commonly used for preventing blood clots. The dataset contains information on the patients, therapeutic doses, and the resulting outcomes. The outcomes of the treatments are reported in the international normalized ratio (INR), which measures how quickly the blood clots. We use a similar experimental setting used by Kallus and Zhou [2]. The 81 features of the patient information selected by Kallus and Zhou [2] is used as states $\mathbf{s}$. For the behavior action vectors, the first dimension is sampled from the normal distribution $N(\mu^* + \sigma^*\sqrt{0.5}z_{BMI}, (\sigma^*\sqrt{0.5})^2)$ truncated by the minimum and maximum therapeutic doses $a^*_{\min}$ and $a^*_{\max}$ in the data. $z_{BMI}$ is the z score of patients' BMIs, $\mu^*$ and $\sigma^*$ are the mean and the standard deviation of the therapeutic dosages, respectively. To test our algorithm we add a dummy action dimension sampled from a uniform distribution $a_2 \sim \mathrm{unif}[a^*_{\min}, a^*_{\max}]$. For the reward, since Kallus and Zhou [2] reported that the INR in the dataset is inadequate for testing OPE algorithms, we use their cost function as a negative reward function $r = -\max(|a_1 - a^*| - 0.1a^*, 0)$. The reward function is only dependent on the first action dimension. The target policy is $\pi(\mathbf{s}) = \begin{bmatrix} s_{BMI} \\ 0 \end{bmatrix}$.

Experimental results in Figure 4a show that our metric reduces the MSE of the kernel-based IS estimate in Warfarin data that is more realistic than that of the synthetic domain. In the figure, we see that even though the MSE of the reward regressor is high, MSE of SLOPE and Kallus and Zhou's estimator is reduced with the KMIS metric. Similar to the results in the synthetic domains, the KMIS metric reduces MSE for almost all given bandwidths in Figure 4b.

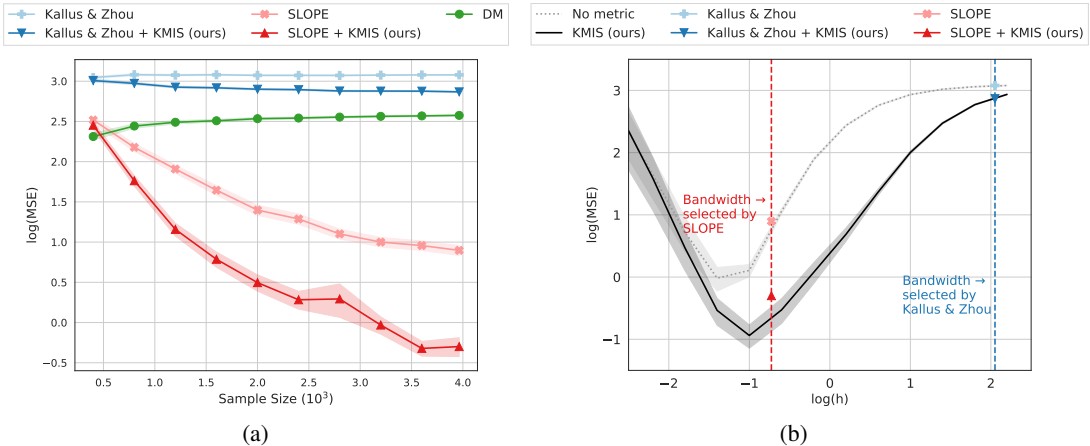

<p style="text-align:center">(a)             (b)</p>

Figure 4: Experimental results on the Warfarin Data. (a) Performance of estimators on the Warfarin Data with increasing data size. The experiment is repeated for 300 trials, and means and standard errors of the squared errors are reported. (b) Reduction of MSEs by metrics at given bandwidths. The experiment is repeated for 100 trials. Means and standard errors of the squared errors are reported. The dotted vertical lines present the average of the bandwidths selected by Kallus and Zhou's estimator (blue) or SLOPE (red), and the markers show the average of the selected bandwidths and the MSEs of each estimator when all 3964 samples are used to obtain the OPE results. Since the selected bandwidth can vary for each run, the markers may not exactly lie on the black lines denoting the fixed bandwidth's results.

# 6 Conclusion

We presented KMIS, an algorithm for off-policy evaluation in contextual bandits for a deterministic target policy with multidimensional continuous action space. KMIS improves the kernel-based IS method by learning a local distance metric used in the kernel function per state, leading to a kernel-based IS OPE estimator that exploits Mahalanobis distance, rather than Euclidean distance. Based on the observation that the leading-order bias becomes a dominant term in the LOMSE when the action dimensionality is high, we derived an analytic solution to the optimal metric matrix that minimizes the upper bound of the leading-order bias. The optimal metric matrix is bandwidth-agnostic and computed using the Hessian of the learned reward function. Experimental results demonstrated that our KMIS significantly improves the performance of the kernel-based IS OPE across different bandwidth selection methods, outperforming baseline algorithms. As for future work, exploring algorithms that jointly optimize bandwidth and metric or extending to RL beyond contextual bandits case would be interesting directions to pursue.

## Acknowledgments and Disclosure of Funding

Haanvid Lee, Yunseon Choi, and Kee-Eung Kim were supported by National Research Foundation (NRF) of Korea (NRF-2019R1A2C1087634), Field-oriented Technology Development Project for Customs Administration through National Research Foundation (NRF) of Korea funded by the Ministry of Science & ICT and Korea Customs Service (NRF-2021M3I1A1097938), Institute of Information & communications Technology Planning & Evaluation (IITP) grant funded by the Korea government (MSIT) (No.2020-0-00940, Foundations of Safe Reinforcement Learning and Its Applications to Natural Language Processing; No.2022-0-00311, Development of Goal-Oriented Reinforcement Learning Techniques for Contact-Rich Robotic Manipulation of Everyday Objects; No.2019-0-00075, Artificial Intelligence Graduate School Program (KAIST); No.2021-0-02068, Artificial Intelligence Innovation Hub), and Electronics and Telecommunications Research Institute (ETRI) grant funded by the Korean government (22ZS1100, Core Technology Research for Self-Improving Integrated Artificial Intelligence System), and KAIST-NAVER Hypercreative AI Center. Byung-Jun Lee was supported by Institute of Information & communications Technology Planning & Evaluation (IITP) grant funded by the Korea government(MSIT) (No.2019-0-00079 , Artificial Intelligence Graduate School Program(Korea University)).

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
