# A Derivation Details

## A.1 Leading-Order MSE

### Bias of the Kernel-Based IS Estimation [2]

Define $\mathbb{E}_{\pi_b}[\cdot] := \mathbb{E}_{\mathbf{s}\sim p(\mathbf{s}),\mathbf{a}\sim \pi_b(\mathbf{a}|\mathbf{s}),r\sim p(r|\mathbf{s},\mathbf{a})}[\cdot]$, then the bias of a kernel-based IS estimate is:

$$\text{Bias}[\hat{\rho}^K] = \mathbb{E}_{\pi_b}[\hat{\rho}^K] - \rho^\pi$$

$$= \mathbb{E}_{\pi_b}\left[\frac{1}{Nh^{D_A}}\sum_{i=1}^{N}K\left(\frac{\mathbf{a}_i - \pi(\mathbf{s}_i)}{h}\right)\frac{r_i}{\pi_b(\mathbf{a}_i\,|\,\mathbf{s}_i)}\right] - \rho^\pi. \tag{12}$$

For the first term of Eq. (12),

$$\mathbb{E}_{\pi_b}\left[\frac{1}{Nh^{D_A}}\sum_{i=1}^{N}K\left(\frac{\mathbf{a}_i - \pi(\mathbf{s}_i)}{h}\right)\frac{r_i}{\pi_b(\mathbf{a}_i\,|\,\mathbf{s}_i)}\right]$$

$$= \frac{1}{h^{D_A}}\iiint p(\mathbf{s})p(r|\mathbf{s},\mathbf{a})K\left(\frac{\mathbf{a}-\pi(\mathbf{s})}{h}\right)r\,dr\,d\mathbf{a}\,d\mathbf{s}$$

$$= \iiint rp(\mathbf{s})\left(p(r|\mathbf{s},\pi(\mathbf{s})) + \frac{h^2}{2}\mathbf{u}^\top\,\mathbf{H}_\mathbf{a}\,p(r|\mathbf{s},\mathbf{a})|_{\mathbf{a}=\pi(\mathbf{s})}\,\mathbf{u} + O\left(h^4\right)\right)K(\mathbf{u})\,d\mathbf{u}\,dr\,d\mathbf{s}$$

$$= \rho^\pi + \frac{h^2}{2}\iint rp(\mathbf{s})\,\text{tr}\left(\left[\int \mathbf{u}\,\mathbf{u}^\top K(\mathbf{u})\,d\mathbf{u}\right]\mathbf{H}_\mathbf{a}\,p(r|\mathbf{s},\mathbf{a})|_{\mathbf{a}=\pi(\mathbf{s})}\right)dr\,d\mathbf{s} + O\left(h^4\right)$$

$$= \rho^\pi + \frac{h^2}{2}\iint rp(\mathbf{s})\,\nabla_\mathbf{a}^2 p(r|\mathbf{s},\mathbf{a})|_{\mathbf{a}=\pi(\mathbf{s})}\,dr\,d\mathbf{s} + O\left(h^4\right),$$

$$\therefore \text{Bias}[\hat{\rho}^K] = \frac{h^2}{2}\iint rp(\mathbf{s})\,\nabla_\mathbf{a}^2 p(r|\mathbf{s},\mathbf{a})|_{\mathbf{a}=\pi(\mathbf{s})}\,dr\,d\mathbf{s} + O\left(h^4\right), \tag{13}$$

where the following relations are used in the derivation:

$$\int K(\mathbf{u})d\mathbf{u} = 1,$$

$$\int \mathbf{u}K(\mathbf{u})d\mathbf{u} = 0,$$

$$\kappa_2(K) := \int \mathbf{u}\mathbf{u}^\top K(\mathbf{u})d\mathbf{u} = I \text{ (By design on } K\text{)},$$

$$\mathbf{u} := \frac{\mathbf{a}-\pi(\mathbf{s})}{h},$$

$$h^{D_A}d\mathbf{u} = d\mathbf{a}.$$

Taylor expansion of $p(r|\mathbf{s},\mathbf{a})$ at $\mathbf{a} = \pi(\mathbf{s})$ is also used for the derivation.

$$p(r|\mathbf{s},\mathbf{a}) = p(r|\mathbf{s},\pi(\mathbf{s})) + (\mathbf{a}-\pi(\mathbf{s}))^\top \nabla_\mathbf{a} p(r|\mathbf{s},\mathbf{a})|_{\mathbf{a}=\pi(\mathbf{s})}$$

$$+ \frac{1}{2}(\mathbf{a}-\pi(\mathbf{s}))^\top \mathbf{H}_\mathbf{a}\,p(r|\mathbf{s},\mathbf{a})|_{\mathbf{a}=\pi(\mathbf{s})}(\mathbf{a}-\pi(\mathbf{s})) + \dots$$

$$= p(r|\mathbf{s},\pi(\mathbf{s})) + h\mathbf{u}^\top \nabla_\mathbf{a} p(r|\mathbf{s},\mathbf{a})|_{\mathbf{a}=\pi(\mathbf{s})}$$

$$+ \frac{h^2}{2}\mathbf{u}^\top \mathbf{H}_\mathbf{a}\,p(r|\mathbf{s},\mathbf{a})|_{\mathbf{a}=\pi(\mathbf{s})}\mathbf{u} + \dots,$$

where in the second equality, $h\mathbf{u} = \mathbf{a} - \pi(\mathbf{s})$ was used.

**Variance of the Kernel-Based IS Estimation [2]**

Define $\mathbb{E}_{\pi_b}[\cdot] := \mathbb{E}_{\mathbf{s}\sim p(\mathbf{s}),\mathbf{a}\sim\pi_b(\mathbf{a}|\mathbf{s}),r\sim p(r|\mathbf{s},\mathbf{a})}[\cdot]$, then the variance of a kernel-based IS estimate is:

$$
\begin{aligned}
\mathrm{Var}[\hat{\rho}^K] &= \mathrm{Var}\left[\frac{1}{Nh^{D_A}}\sum_{i=1}^{N}K\left(\frac{\mathbf{a}_i-\pi(\mathbf{s}_i)}{h}\right)\frac{r_i}{\pi_b(\mathbf{a}_i\mid\mathbf{s}_i)}\right]\\
&= \frac{1}{N^2}\times N\times\mathrm{Var}\left[\frac{1}{h^{D_A}}K\left(\frac{\mathbf{a}-\pi(\mathbf{s})}{h}\right)\frac{r}{\pi_b(\mathbf{a}\mid\mathbf{s})}\right]\\
&= \frac{1}{N}\left\{\mathbb{E}_{\pi_b}\left[\left(\frac{1}{h^{D_A}}K\left(\frac{\mathbf{a}-\pi(\mathbf{s})}{h}\right)\frac{r}{\pi_b(\mathbf{a}\mid\mathbf{s})}\right)^2\right]\right.\\
&\qquad\qquad \left.-\left(\mathbb{E}_{\pi_b}\left[\frac{1}{h^{D_A}}K\left(\frac{\mathbf{a}-\pi(\mathbf{s})}{h}\right)\frac{r}{\pi_b(\mathbf{a}\mid\mathbf{s})}\right]\right)^2\right\}.
\end{aligned}
\tag{14}
$$

The second term of the Eq. (14) is,

$$
\begin{aligned}
&\frac{1}{N}\left(\mathbb{E}_{\pi_b}\left[\frac{1}{h^{D_A}}K\left(\frac{\mathbf{a}-\pi(\mathbf{s})}{h}\right)\frac{r}{\pi_b(\mathbf{a}\mid\mathbf{s})}\right]\right)^2\\
&= \frac{1}{N}\left(\mathrm{Bias}[\hat{\rho}^K]+\rho^\pi\right)^2\\
&= \frac{1}{N}\left[\rho^\pi+\frac{h^2}{2}\iint rp(\mathbf{s})\left.\nabla_{\mathbf{a}}^2 p(r|\mathbf{s},\mathbf{a})\right|_{\mathbf{a}=\pi(\mathbf{s})}dr\,d\mathbf{s}+O\left(h^4\right)\right]^2,\\
&\therefore \frac{1}{N}\left(\mathbb{E}_{\pi_b}\left[\frac{1}{h^{D_A}}K\left(\frac{\mathbf{a}-\pi(\mathbf{s})}{h}\right)\frac{r}{\pi_b(\mathbf{a}\mid\mathbf{s})}\right]\right)^2 = O\left(\frac{1}{N}\right).
\end{aligned}
\tag{15}
$$

The first term of the Eq. (14) is,

$$
\begin{aligned}
&\frac{1}{N}\mathbb{E}_{\pi_b}\left[\left(\frac{1}{h^{D_A}}K\left(\frac{\mathbf{a}-\pi(\mathbf{s})}{h}\right)\frac{r}{\pi_b(\mathbf{a}\mid\mathbf{s})}\right)^2\right]\\
&= \frac{1}{N}\mathbb{E}_{\mathbf{s}\sim p(\mathbf{s})}\left[\iint r^2\frac{p(r|\mathbf{s},\mathbf{a})}{h^{2D_A}\pi_b(\mathbf{a}\mid\mathbf{s})}K\left(\frac{\mathbf{a}-\pi(\mathbf{s})}{h}\right)^2 d\mathbf{a}\,dr\right]\\
&= \frac{1}{N}\mathbb{E}_{\mathbf{s}\sim p(\mathbf{s})}\left[\iint r^2\frac{K(\mathbf{u})^2}{h^{D_A}}\frac{p(r|\mathbf{s},h\mathbf{u}+\pi(\mathbf{s}))}{\pi_b(h\mathbf{u}+\pi(\mathbf{s})\mid\mathbf{s})}d\mathbf{u}\,dr\right],
\end{aligned}
\tag{16}
$$

where in the second equality, $\mathbf{a}=h\mathbf{u}+\pi(\mathbf{s})$ was used (which comes from $\mathbf{u}=\frac{\mathbf{a}-\pi(\mathbf{s})}{h}$).

Let $g_r(\mathbf{s},\mathbf{a}):=\frac{p(r|\mathbf{s},\mathbf{a})}{\pi_b(\mathbf{a}\mid\mathbf{s})}$, and apply Taylor expansion at $\mathbf{a}=\pi(\mathbf{s})$,

$$
\begin{aligned}
g_r(\mathbf{s},\mathbf{a}) = g_r(\mathbf{s},\pi(\mathbf{s})) &+ (\mathbf{a}-\pi(\mathbf{s}))^\top\left.\nabla_{\mathbf{a}}g_r(\mathbf{s},\mathbf{a})\right|_{\mathbf{a}=\pi(\mathbf{s})}\\
&+ \frac{1}{2}(\mathbf{a}-\pi(\mathbf{s}))^\top\left.\mathbf{H}_{\mathbf{a}}\,g_r(\mathbf{s},\mathbf{a})\right|_{\mathbf{a}=\pi(\mathbf{s})}(\mathbf{a}-\pi(\mathbf{s}))+O(h^3).
\end{aligned}
$$

By using $\mathbf{a} = h\,\mathbf{u} + \pi(\mathbf{s})$,

$$g_r(\mathbf{s}, h\,\mathbf{u} + \pi(\mathbf{s})) = g_r(\mathbf{s}, \pi(\mathbf{s})) + h\mathbf{u}^\top \nabla_a g_r(\mathbf{s}, \mathbf{a})\,|_{\mathbf{a}=\pi(\mathbf{s})} \tag{17}$$
$$+ \frac{h^2}{2}\mathbf{u}^\top \mathbf{H_a}\, g_r(\mathbf{s}, \mathbf{a})\,|_{\mathbf{a}=\pi(\mathbf{s})}\,\mathbf{u} + O(h^3).$$

By plugging in Eq. (17) to Eq. (16),

$$\frac{1}{N}\mathbb{E}_{\mathbf{s}\sim p(\mathbf{s})}\left[\iint r^2 \frac{K(\mathbf{u})^2}{h^{D_A}} \frac{p(r\,|\,\mathbf{s}, h\,\mathbf{u}+\pi(\mathbf{s}))}{\pi_b(h\,\mathbf{u}+\pi(\mathbf{s})\,|\,\mathbf{s})}d\,\mathbf{u}\,dr\right]$$

$$= \frac{1}{Nh^{D_A}}\mathbb{E}_{\mathbf{s}\sim p(\mathbf{s})}\left[\iint r^2 K(\mathbf{u})^2\left(g_r(\mathbf{s}, \pi(\mathbf{s})) + \frac{h^2}{2}\mathbf{u}^\top \mathbf{H_a}\, g_r(\mathbf{s}, \mathbf{a})\,|_{\mathbf{a}=\pi(\mathbf{s})}\,\mathbf{u} + O(h^4)\right)d\,\mathbf{u}\,dr\right]$$

$$= \frac{R(K)}{Nh^{D_A}}\mathbb{E}_{\mathbf{s}\sim p(\mathbf{s})}\left[\int \frac{r^2 p(r\,|\,\mathbf{s}, \mathbf{a}=\pi(\mathbf{s}))}{\pi_b(\mathbf{a}=\pi(\mathbf{s})\,|\,\mathbf{s})}dr\right]$$

$$+ \frac{1}{2Nh^{D_A-2}}\mathbb{E}_{\mathbf{s}\sim p(\mathbf{s})}\left[\iint r^2 K(\mathbf{u})^2\left(\mathbf{u}^\top \mathbf{H_a}\, g_r(\mathbf{s}, \mathbf{a})\,|_{\mathbf{a}=\pi(\mathbf{s})}\,\mathbf{u} + O(h^4)\right)d\,\mathbf{u}\,dr\right]$$

$$= \frac{R(K)}{Nh^{D_A}}\mathbb{E}_{\mathbf{s}\sim p(\mathbf{s})}\left[\int \frac{r^2 p(r\,|\,\mathbf{s}, \mathbf{a}=\pi(\mathbf{s}))}{\pi_b(\mathbf{a}=\pi(\mathbf{s})\,|\,\mathbf{s})}dr\right] + O\left(\frac{1}{Nh^{D_A-2}}\right), \tag{18}$$

where $R(K) := \int K(\mathbf{u})^2 d\,\mathbf{u}$.

From Eq. (15) and Eq. (18), the variance can be represented as,

$$\therefore \mathrm{Var}[\hat{\rho}^K] = \frac{R(K)}{Nh^{D_A}}\mathbb{E}_{\mathbf{s}\sim p(\mathbf{s})}\left[\frac{\mathbb{E}\left[r^2\,|\,\mathbf{s}, \mathbf{a}=\pi(\mathbf{s})\right]}{\pi_b(\mathbf{a}=\pi(\mathbf{s})\,|\,\mathbf{s})}\right] + O\left(\frac{1}{Nh^{D_A-2}}\right). \tag{19}$$

From the derived bias (Eq. (13)) and variance (Eq. (19)), MSE can be derived:

$$\mathrm{MSE}[\hat{\rho}^K] = \mathrm{Bias}[\hat{\rho}^K]^2 + \mathrm{Var}[\hat{\rho}^K]$$

$$= \left(\frac{h^2}{2}\iint rp(\mathbf{s})\,\nabla_\mathbf{a}^2 p(r|\mathbf{s}, \mathbf{a})\big|_{\mathbf{a}=\pi(\mathbf{s})}\,dr\,d\mathbf{s}\right)^2 + O\left(h^6\right)$$

$$+ \frac{R(K)}{Nh^{D_A}}\mathbb{E}_{\mathbf{s}\sim p(\mathbf{s})}\left[\frac{\mathbb{E}\left[r^2\,|\,\mathbf{s}, \mathbf{a}=\pi(\mathbf{s})\right]}{\pi_b(\mathbf{a}=\pi(\mathbf{s})\,|\,\mathbf{s})}\right] + O\left(\frac{1}{Nh^{D_A-2}}\right).$$

Assuming that $h \to 0$ and $\frac{1}{Nh^{D_A}} \to 0$ as $N \to \infty$,

$$\mathrm{MSE}[\hat{\rho}^K] \approx h^4 C_b + \frac{C_v}{Nh^{D_A}}$$

$$=: \mathrm{LOMSE}(h, N, D_A),$$

$$C_b := \left(\frac{1}{2}\mathbb{E}_{\mathbf{s}\sim p(\mathbf{s})}\left[\nabla_\mathbf{a}^2\left(\mathbb{E}[r\,|\,\mathbf{s}, \mathbf{a}]\right)\big|_{\mathbf{a}=\pi(\mathbf{s})}\right]\right)^2, C_v := R(K)\mathbb{E}_{\mathbf{s}\sim p(\mathbf{s})}\left[\frac{\mathbb{E}[r^2\,|\,\mathbf{s}, \mathbf{a}=\pi(\mathbf{s})]}{\pi_b(\mathbf{a}=\pi(\mathbf{s})\,|\,\mathbf{s})}\right].$$

## A.2 Optimal Bandwidth

The optimal bandwidth $(h^*)$ that minimizes the leading-order MSE [2] is,

$$\frac{d}{dh}(\mathrm{LOMSE}(h, N, D_A)) = 4h^3 C_b - D_A h^{-D_A-1} N^{-1} C_v,$$

$$4(h^*)^3 C_b - D_A (h^*)^{-D_A-1} N^{-1} C_v = 0,$$

$$h^* = \left(\frac{D_A C_v}{4N C_b}\right)^{\frac{1}{D_A+4}}.$$

## A.3 Derivation of Eq. (6)

For the derivation, we use the following relations:

$$p(r \mid \mathbf{s}, \mathbf{z}) = \frac{p(r, \mathbf{s}, \mathbf{z})}{p(\mathbf{s}, \mathbf{z})} \tag{20}$$

$$= \frac{p(r, \mathbf{s}, \mathbf{a}) \left|\frac{\partial \mathbf{a}}{\partial \mathbf{z}}\right|}{p(\mathbf{s}, \mathbf{a}) \left|\frac{\partial \mathbf{a}}{\partial \mathbf{z}}\right|}$$

$$= p(r \mid \mathbf{s}, \mathbf{a}),$$

$$\mathbf{H}_{\mathbf{z}} \, p(r \mid \mathbf{s}, \mathbf{z})\big|_{\mathbf{z}=L(\mathbf{s})^\top \pi(\mathbf{s})} = \frac{\partial^2}{\partial \mathbf{z} \, \partial \mathbf{z}^\top} p(r \mid \mathbf{s}, \mathbf{z}) \tag{21}$$

$$= \frac{\partial}{\partial \mathbf{z}} \left(\frac{\partial}{\partial \mathbf{z}} p(r \mid \mathbf{s}, \mathbf{z})\right)^\top$$

$$= \frac{\partial}{\partial \mathbf{z}} \left(L(\mathbf{s})^{-1} \frac{\partial}{\partial \mathbf{a}} p(r \mid \mathbf{s}, \mathbf{a})\right)^\top, \text{ by Eq. (20)}$$

$$= \frac{\partial \mathbf{a}}{\partial \mathbf{z}} \frac{\partial}{\partial \mathbf{a}} \left(\left(\frac{\partial}{\partial \mathbf{a}} p(r \mid \mathbf{s}, \mathbf{a})\right)^\top L(\mathbf{s})^{-\top}\right)$$

$$= L(\mathbf{s})^{-1} \frac{\partial^2}{\partial \mathbf{a} \, \partial \mathbf{a}^\top} p(r \mid \mathbf{s}, \mathbf{a}) L(\mathbf{s})^{-\top}$$

$$= L(\mathbf{s})^{-1} \mathbf{H}_{\mathbf{a}} \, p(r \mid \mathbf{s}, \mathbf{a})\big|_{\mathbf{a}=\pi(\mathbf{s})} L^{-\top}(\mathbf{s}).$$

By changing $\mathbf{a}$ of $C_b$ in Eq. (2) to $\mathbf{z}$ $(= L(\mathbf{s})^\top \mathbf{a})$, we get:

$$C_{b,A} = \left(\frac{1}{2} \mathbb{E}_{\mathbf{s}\sim p(\mathbf{s})} \left[\nabla_{\mathbf{z}}^2 r\,(\mathbf{s}, \mathbf{z})\big|_{\mathbf{z}=L(\mathbf{s})^\top \pi(\mathbf{s})}\right]\right)^2$$

$$= \left(\frac{1}{2} \iint r p(\mathbf{s}) \,\mathrm{tr}\left(\mathbf{H}_{\mathbf{z}} \, p(r \mid \mathbf{s}, \mathbf{z})\big|_{\mathbf{z}=L(\mathbf{s})^\top \pi(\mathbf{s})}\right) dr d\mathbf{s}\right)^2$$

$$= \left(\frac{1}{2} \mathbb{E}_{\mathbf{s}\sim p(\mathbf{s})} \left[\mathrm{tr}\left(L(\mathbf{s})^{-1} \mathbf{H}_{\mathbf{a}} \, r(\mathbf{s}, \mathbf{a})\big|_{\mathbf{a}=\pi(\mathbf{s})} L(\mathbf{s})^{-\top}\right)\right]\right)^2, \text{ by Eq. (21)}$$

$$= \left(\frac{1}{2} \mathbb{E}_{\mathbf{s}\sim p(\mathbf{s})} \left[\mathrm{tr}\left((L(\mathbf{s})L(\mathbf{s})^\top)^{-1} \mathbf{H}_{\mathbf{a}} \, r(\mathbf{s}, \mathbf{a})\big|_{\mathbf{a}=\pi(\mathbf{s})}\right)\right]\right)^2,$$

$$\therefore C_{b,A} = \left(\frac{1}{2} \mathbb{E}_{\mathbf{s}\sim p(\mathbf{s})} \left[\mathrm{tr}\left(A(\mathbf{s})^{-1} \mathbf{H}_{\mathbf{a}} \, r(\mathbf{s}, \mathbf{a})\big|_{\mathbf{a}=\pi(\mathbf{s})}\right)\right]\right)^2.$$

## B Proofs

### B.1 Proof of Proposition 1

**Proposition 1.** *(Adapted from Noh et al. [23]) For a high dimensional action space $D_A \gg 4$, and given optimal bandwidth $h^*$ (Eq. (3)), the squared leading-order bias dominates over the leading-order variance in LOMSE. Furthermore,* $\mathrm{LOMSE}(h^*, N, D_A)$ *can be approximated by $C_b$ in Eq. (2).*

$$\mathrm{LOMSE}(h^*, N, D_A) = N^{-\frac{4}{D_A+4}} \left( \left( \frac{D_A}{4} \right)^{\frac{4}{D_A+4}} + \left( \frac{4}{D_A} \right)^{\frac{D_A}{D_A+4}} \right) C_b^{\frac{D_A}{D_A+4}} C_v^{\frac{4}{D_A+4}} \approx C_b. \quad (22)$$

*Proof.* By plugging in $h^*$ in Eq. (3) to the leading-order bias and leading-order variance in Eq. (2) and by taking $D_A \to \infty$ (or, for $D_A \gg 4$), their ratio is:

$$\lim_{D_A \to \infty} \frac{(\text{leading-order bias})^2}{\text{leading-order var}} = \lim_{D_A \to \infty} \left( \frac{D_A}{4} \right) = \infty. \quad (23)$$

Therefore, the squared leading-order bias dominates over the leading-order variance in the LOMSE in the high dimensional action space ($D_A \gg 4$).

By plugging in $h^*$ Eq. (3) to the LOMSE in Eq. (2) we get,

$$\mathrm{LOMSE}(h^*, N, D_A) = N^{-\frac{4}{D_A+4}} \left( \left( \frac{D_A}{4} \right)^{\frac{4}{D_A+4}} + \left( \frac{4}{D_A} \right)^{\frac{D_A}{D_A+4}} \right) C_b^{\frac{D_A}{D_A+4}} C_v^{\frac{4}{D_A+4}}.$$

By taking $D_A \to \infty$ (or, for $D_A \gg 4$),

$$\lim_{D_A \to \infty} N^{-\frac{4}{D_A+4}} = 1,$$

$$\lim_{D_A \to \infty} \left( \frac{4}{D_A} \right)^{\frac{4}{D_A+4}} = 0,$$

$$\lim_{D_A \to \infty} C_b^{\frac{D_A}{D_A+4}} = C_b,$$

$$\lim_{D_A \to \infty} C_v^{\frac{4}{D_A+4}} = 1,$$

$$\lim_{D_A \to \infty} \left( \frac{D_A}{4} \right)^{\frac{4}{D_A+4}} = 1,$$

$$\therefore \mathrm{LOMSE}(h^*, N, D_A) = N^{-\frac{4}{D_A+4}} \left( \left( \frac{D_A}{4} \right)^{\frac{4}{D_A+4}} + \left( \frac{4}{D_A} \right)^{\frac{D_A}{D_A+4}} \right) C_b^{\frac{D_A}{D_A+4}} C_v^{\frac{4}{D_A+4}}$$

$$\approx C_b. \text{ (for } D_A \gg 4).$$

$\square$

## B.2 Proof of Theorem 1

We use the semi-definite programming solution presented in the work of Noh et al. [22] for computing the metric matrix $A^*(\mathbf{s})$ that minimizes the $\text{tr}\left(A^{-1}(\mathbf{s})B(\mathbf{s})\right)^2$ for nearest neighbor classification. We use the solution to minimize the squared trace term in Eq. (7). In our case, we use $B(\mathbf{s}) := \mathbf{H_a}\,\mathbb{E}[r\,|\,\mathbf{s},\mathbf{a}]|_{\mathbf{a}=\pi(\mathbf{s})}$ which comes from Eq. (7).

**Theorem 1.** *(Adapted from Noh et al. [22]) Assume that the $p(r\,|\,\mathbf{s},\mathbf{a})$ is twice differentiable w.r.t. an action $\mathbf{a}$. Let $\Lambda_+(\mathbf{s})$ and $\Lambda_-(\mathbf{s})$ be diagonal matrices of positive and negative eigenvalues of the Hessian $\mathbf{H_a}\,\mathbb{E}[r\,|\,\mathbf{s},\mathbf{a}]|_{\mathbf{a}=\pi(\mathbf{s})}$, $U_+(\mathbf{s})$ and $U_-(\mathbf{s})$ be matrices of eigenvectors corresponding to $\Lambda_+(\mathbf{s})$ and $\Lambda_-(\mathbf{s})$ respectively, and $d_+(\mathbf{s})$ and $d_-(\mathbf{s})$ be the numbers of positive and negative eigenvalues of the Hessian. Then the metric $A^*(\mathbf{s})$ that minimizes $U_{b,A}$ is:*

$$A^*(\mathbf{s}) = \alpha(\mathbf{s})\,[U_+(\mathbf{s})U_-(\mathbf{s})] \underbrace{\begin{pmatrix} d_+(\mathbf{s})\Lambda_+(\mathbf{s}) & 0 \\ 0 & -d_-(\mathbf{s})\Lambda_-(\mathbf{s}) \end{pmatrix}}_{=:M(\mathbf{s})} [U_+(\mathbf{s})U_-(\mathbf{s})]^\top, \qquad (24)$$

*where $\alpha(\mathbf{s}) := |M(\mathbf{s})|^{-1/(d_+(\mathbf{s})+d_-(\mathbf{s}))}$.*

*Proof.* Define $B(\mathbf{s}) := \mathbf{H_a}\,r(\mathbf{s},\mathbf{a})|_{\pi(\mathbf{s})}$, then we want $A(\mathbf{s})$ that,

$$\min_{A(\mathbf{s})} \left(\text{tr}\left(A(\mathbf{s})^{-1}B(\mathbf{s})\right)\right)^2 \qquad (25)$$
$$\text{s.t. } |A(\mathbf{s})| = 1,$$
$$A(\mathbf{s}) \succ 0,$$
$$A(\mathbf{s}) = A(\mathbf{s})^\top.$$

In the cases where some of the eigenvalues of $B(\mathbf{s})$ are zero, the components along the directions of the corresponding eigenvectors can be discarded.

Minimizing $U_{b,A}$ is the same as minimizing $C_{b,A}$ under the condition that there are not both sets of states with positive and negative trace terms $\text{tr}\left(A(\mathbf{s})^{-1}B(\mathbf{s})\right)$. The trace term is positive or negative when the eigenvalues of $B(\mathbf{s})$ other than zero are all positive or negative, respectively. And the trace term becomes zero when there are both negative and positive eigenvalues in the eigenvalues of $B(\mathbf{s})$. In cases where all the states have both positive and negative eigenvalues in the eigenvalues of $B(\mathbf{s})$, $C_{b,A} = U_{b,A} = 0$.

When there are both positive and negative eigenvalues in the eigenvalues of $B(\mathbf{s})$,

$$A^*(\mathbf{s}) = \alpha(\mathbf{s})\,[U_+(\mathbf{s})U_-(\mathbf{s})] \begin{pmatrix} d_+(\mathbf{s})\Lambda_+(\mathbf{s}) & 0 \\ 0 & -d_-(\mathbf{s})\Lambda_-(\mathbf{s}) \end{pmatrix} [U_+(\mathbf{s})U_-(\mathbf{s})]^\top,$$

$$B(\mathbf{s}) = [U_+(\mathbf{s})U_-(\mathbf{s})] \begin{pmatrix} \Lambda_+(\mathbf{s}) & 0 \\ 0 & \Lambda_-(\mathbf{s}) \end{pmatrix} [U_+(\mathbf{s})U_-(\mathbf{s})]^\top,$$

$$\text{tr}\left(A^*(\mathbf{s})^{-1}B(\mathbf{s})\right)$$
$$= \frac{1}{\alpha(\mathbf{s})}\,\text{tr}\left[[U_+(\mathbf{s})U_-(\mathbf{s})] \begin{pmatrix} \frac{\Lambda_+^{-1}(\mathbf{s})}{d_+(\mathbf{s})} & 0 \\ 0 & -\frac{\Lambda_-^{-1}(\mathbf{s})}{d_-(\mathbf{s})} \end{pmatrix} \begin{pmatrix} \Lambda_+(\mathbf{s}) & 0 \\ 0 & \Lambda_-(\mathbf{s}) \end{pmatrix} [U_+(\mathbf{s})U_-(\mathbf{s})]^\top\right]$$
$$= \frac{1}{\alpha(\mathbf{s})}\left(\frac{d_+(\mathbf{s})}{d_+(\mathbf{s})} - \frac{d_-(\mathbf{s})}{d_-(\mathbf{s})}\right)$$
$$= 0.$$

When the eigenvalues of $B(\mathbf{s})$ other than zero are all negative or positive, then we can solve the Lagrangian equation $F(\mathbf{s})$ in Eq. (26). We first solve for the case when the eigenvalues of $B(\mathbf{s})$ other than zero are all positive,

$$F(\mathbf{s}) = \mathrm{tr}\left(A(\mathbf{s})^{-1}B(\mathbf{s})\right)^2 - c(\mathbf{s})(|A(\mathbf{s})| - 1), \tag{26}$$

$$\frac{\partial F(\mathbf{s})}{\partial c(\mathbf{s})} = |A(\mathbf{s})| - 1 = 0,$$

$$\frac{\partial F(\mathbf{s})}{\partial A(\mathbf{s})} = -2\,\mathrm{tr}\left(A(\mathbf{s})^{-1}B(\mathbf{s})\right)A(\mathbf{s})^{-\top}B(\mathbf{s})^{\top}A(\mathbf{s})^{-\top} - c(\mathbf{s})|A(\mathbf{s})|A(\mathbf{s})^{-\top} = 0. \tag{27}$$

From Eq. (27),

$$c(\mathbf{s})I = -2\,\mathrm{tr}\left(A(\mathbf{s})^{-1}B(\mathbf{s})\right)A(\mathbf{s})^{-1}B(\mathbf{s}). \tag{28}$$

$$\left(\text{since } A(\mathbf{s}) = A(\mathbf{s})^{\top},\ B(\mathbf{s}) = B(\mathbf{s})^{\top},\ |A(\mathbf{s})| = 1\right)$$

(i) Assume $c(\mathbf{s}) = 0$ in Eq. (28),

Then either $\mathrm{tr}\left(A(\mathbf{s})^{-1}B(\mathbf{s})\right) = 0$, or, $A(\mathbf{s})^{-1}B(\mathbf{s}) = \mathbf{0}$.

We first check if $\mathrm{tr}\left(A(\mathbf{s})^{-1}B(\mathbf{s})\right) = 0$.

$$\mathrm{tr}\left(A(\mathbf{s})^{-1}B(\mathbf{s})\right) = \mathrm{tr}\left(B(\mathbf{s})^{\frac{1}{2}}A(\mathbf{s})^{-1}B(\mathbf{s})^{\frac{\top}{2}}\right). \tag{29}$$

Since $B(\mathbf{s}) = U_+(\mathbf{s})\Lambda_+(\mathbf{s})U_+(\mathbf{s})^{\top}$, $B(\mathbf{s})^{\frac{1}{2}} = B(\mathbf{s})^{\frac{\top}{2}} = U_+(\mathbf{s})\Lambda_+(\mathbf{s})^{\frac{1}{2}}U_+(\mathbf{s})^{\top}$. And as $A(\mathbf{s}) \succ 0$, its inverse is also positive definite $A(\mathbf{s})^{-1} \succ 0$. Using these relations, we can show that the term inside the trace in Eq. (29) is a positive definite matrix.

$$\mathbf{a}^{\top}\left[B(\mathbf{s})^{\frac{1}{2}}A(\mathbf{s})^{-1}B(\mathbf{s})^{\frac{\top}{2}}\right]\mathbf{a} = \left(B(\mathbf{s})^{\frac{\top}{2}}\mathbf{a}\right)^{\top}A(\mathbf{s})^{-1}\left(B(\mathbf{s})^{\frac{\top}{2}}\mathbf{a}\right)$$

$$> 0,\ \ \forall\,\mathbf{a} \in \mathbb{R}^{D_A}\backslash\{\mathbf{0}\}.$$

Therefore $B(\mathbf{s})^{\frac{1}{2}}A(\mathbf{s})^{-1}B(\mathbf{s})^{\frac{\top}{2}} \succ 0$ and $\mathrm{tr}\left(B(\mathbf{s})^{\frac{1}{2}}A(\mathbf{s})^{-1}B(\mathbf{s})^{\frac{\top}{2}}\right) > 0$. Then, from Eq. (29),

$$\mathrm{tr}\left(A(\mathbf{s})^{-1}B(\mathbf{s})\right) > 0,$$

$$A(\mathbf{s})^{-1}B(\mathbf{s}) \neq \mathbf{0}, \quad \text{as it's trace is positive.}$$

Therefore, the assumption is wrong, and $c(\mathbf{s}) \neq 0$.

(ii) Assume $c(\mathbf{s}) \neq 0$,

Since the Lagrangian multiplier $c(\mathbf{s})$ and $\mathrm{tr}\left(A(\mathbf{s})^{-1}B(\mathbf{s})\right)$ in Eq. (28) are scalar values, $A(\mathbf{s})$ needs to be a scalar multiple of $B(\mathbf{s})$ (which is symmetric) to match the scalar multiple of identity matrix in the LHS of Eq. (28) while satisfying $A(\mathbf{s}) \succ 0$ and $|A(\mathbf{s})| = 1$. Therefore the $A^*(\mathbf{s})$ is,

$$\therefore A^*(\mathbf{s}) = \alpha(\mathbf{s})U_+(\mathbf{s})\left[d_+(\mathbf{s})\Lambda_+(\mathbf{s})\right]U_+(\mathbf{s})^{\top}. \tag{30}$$

Similarly, we can also derive $A^*(\mathbf{s})$ for the case where all eigenvalues of $B(\mathbf{s})$ are negative. $\qquad\square$

## B.3   Proof of Theorem 2

**Theorem 2.** *(Adapted from Kallus and Zhou [2]) Kernel-based IS estimator with the optimal metric $A^*(\mathbf{s})$ from Eq. (8) and the optimal bandwidth $h^*$ in Eq. (3) is a consistent estimator in which convergence rate is faster than or equal to that of the isotropic kernel-based IS estimator with $h^*$. When the Hessian $\mathbf{H_a}\, r(\mathbf{s}, \mathbf{a})|_{\mathbf{a}=\pi(\mathbf{s})}$ has both positive and negative eigenvalues for all states, $A^*(\mathbf{s})$ applied estimator converges to the true policy value faster than the one without the metric by the rate of $\mathcal{O}(D_A^{-\frac{1}{2}})$ as the action dimension increases.*

*Proof.* The MSE of a kernel-based IS estimation without a metric derived by Kallus and Zhou [2] in terms of bandwidth $h$, data size of $N$, and action dimension $D_A$ is (derivation in Appendix A.1),

$$\text{MSE}(h, N, D_A) = \underbrace{h^4 C_b + O\left(h^6\right)}_{\text{Bias}[\hat{\rho}^K]^2} + \underbrace{\frac{C_v}{Nh^{D_A}} + O\left(\frac{1}{Nh^{D_A-2}}\right)}_{\text{Var}[\hat{\rho}^K]}, \tag{31}$$

where the MSE consists of squared bias and the variance of the estimate $\hat{\rho}^K$. Since the optimal bandwidth presented in Eq. (3) is $h^* = \mathcal{O}\left(\left(\frac{D_A}{N}\right)^{\frac{1}{D_A+4}}\right)$, and $N \gg D_A$, the MSE with $h^*$ and without a metric is as follows:

$$\text{MSE}(h^*, N, D_A) = \mathcal{O}\left(\left(\frac{D_A}{N}\right)^{\frac{4}{D_A+4}}\right) + \mathcal{O}\left(N^{-1}\left(\frac{D_A}{N}\right)^{\frac{-D_A}{D_A+4}}\right) \tag{32}$$

$$= \mathcal{O}\left(\left(\frac{D_A}{N}\right)^{\frac{4}{D_A+4}}\right) + \mathcal{O}\left(\left(\frac{D_A}{N}\right)^{\frac{4}{D_A+4}} \frac{1}{D_A}\right) \tag{33}$$

$$= \mathcal{O}\left(\left(\frac{D_A}{N}\right)^{\frac{4}{D_A+4}}\right). \tag{34}$$

Since the MSE of the kernel-based IS estimator with $h^*$ converges to zero by $\mathcal{O}\left(\left(\frac{D_A}{N}\right)^{\frac{4}{D_A+4}}\right)$, The estimation approaches to the true policy value by the rate of $\mathcal{O}\left(\left(\frac{D_A}{N}\right)^{\frac{2}{D_A+4}}\right)$.

For the optimal metric applied kernel-based IS estimator, its convergence rate can be computed similarly to that of the isotropic kernel-based IS when $C_{b,A^*} \neq 0$ ($C_{b,A^*}$ is $C_{b,A}$ with $A = A^*$), then the resulting convergence rate is the same as that of the isotropic kernel-based IS estimator.

However, in the best case where the Hessians $\mathbf{H_a}\, r(\mathbf{s}, \mathbf{a})|_{\mathbf{a}=\pi(\mathbf{s})}$ contain both negative and positive eigenvalues for all states, $C_{b,A^*} = 0$. Then, the MSE with the optimal metric and the optimal bandwidth $\text{MSE}(h^*, A^*, N, D_A)$ converges to zero by:

$$\text{MSE}(h^*, A^*, N, D_A) = (h^*)^4 \cancel{C_{b,A^*}} + O\left((h^*)^6\right) + \frac{C_v}{N(h^*)^{D_A}} + O\left(\frac{1}{N(h^*)^{D_A-2}}\right) \tag{35}$$

$$= \mathcal{O}\left(\left(\frac{D_A}{N}\right)^{\frac{6}{D_A+4}}\right) + \mathcal{O}\left(\left(\frac{D_A}{N}\right)^{\frac{4}{D_A+4}} \frac{1}{D_A}\right) \tag{36}$$

$$= \mathcal{O}\left(\left(\frac{D_A}{N}\right)^{\frac{4}{D_A+4}} \frac{1}{D_A}\right), \tag{37}$$

where $C_v$ does not change by applying the optimal metric to a kernel due to the constraint $|A^*(\mathbf{s})| = 1$. In the best case where there are both positive and negative eigenvalues for all states, the MSE of our

metric applied kernel-based IS estimation converges to zero faster than the one without the metric by the rate of $\mathcal{O}(D_A^{-1})$:

$$\frac{\mathrm{MSE}(h^*, A^*, N, D_A)}{\mathrm{MSE}(h^*, N, D_A)} = \mathcal{O}\left(\frac{1}{D_A}\right). \tag{38}$$

Therefore, our algorithm converges to the true policy value faster than the one without the metric by the rate of $\mathcal{O}(D_A^{-\frac{1}{2}})$ in the best case. $\qquad\square$

## C  Algorithm Details

### C.1  Regularizers for the KMIS Metric

To compute the regularizers $\beta(\mathbf{s})$ and $\gamma(\mathbf{s})$ in Eq. (10), we first compute $Y(\mathbf{s})$ in Eq. (40) by adding a small positive real coefficient $\epsilon(\mathbf{s})$ to $X(\mathbf{s})$ in Eq. (39).

$$X(\mathbf{s}) := [U_+(\mathbf{s})U_-(\mathbf{s})U_0(\mathbf{s})] \begin{pmatrix} d_+(\mathbf{s})\Lambda_+(\mathbf{s}) & 0 & 0 \\ 0 & -d_-(\mathbf{s})\Lambda_-(\mathbf{s}) & 0 \\ 0 & 0 & \mathbf{0} \end{pmatrix} [U_+(\mathbf{s})U_-(\mathbf{s})U_0(\mathbf{s})]^\top, \tag{39}$$

$$Y(\mathbf{s}) := X(\mathbf{s}) + \epsilon(\mathbf{s})I. \tag{40}$$

We designed $\epsilon(\mathbf{s})$ to be relatively smaller than the eigenvalues of $\frac{A^*(\mathbf{s})}{\alpha(\mathbf{s})}$ (Eq. (8)). For the experiments, $\epsilon(\mathbf{s})$ was assigned to be the maximum absolute value among eigenvalues of the Hessian $\mathbf{H_a}\, r(\mathbf{s}, \mathbf{a})|_{\mathbf{a}=\pi(\mathbf{s})}$ multiplied by 0.01.

When $Y(\mathbf{s})$ is scaled to have determinant of one, then it becomes $\hat{A}(\mathbf{s})$. We scale $Y(\mathbf{s})$ by multiplying $\beta(\mathbf{s}) = |Y(\mathbf{s})|^{\frac{-1}{D_A}}$. Then, $\gamma(\mathbf{s}) = \beta(\mathbf{s})\epsilon(\mathbf{s})$.

## D  Additional Experiment

### D.1  Experiment with Various Noise Levels in the Rewards

When the noise in the rewards of the quadratic reward domain increases, DM suffers the most as the function estimation becomes more difficult as the noise increases. Other algorithms also show an increase in MSEs as the noise increases. The KMIS metric applied kernel-based IS estimator performs the best with the learned metric even though the DM, which the KMIS uses to estimate the Hessians required for the metric learning, performs the worst.

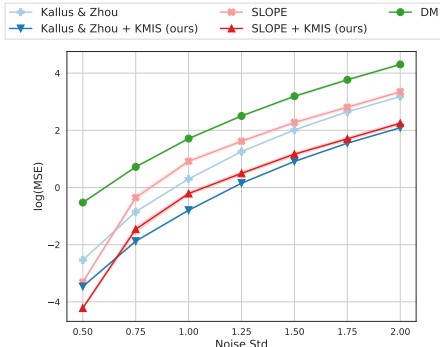

Figure 5: Performance of OPE algorithms in the modified quadratic reward domain with various standard deviations of the Gaussian noise in the rewards. Means and standard errors of squared errors were obtained from 100 trials with 40k samples.

# E    Experiment Details

For all experiments, we used a neural network reward regressor with 2 fully-connected layers of 128 hidden units with tanh activations, and a Gaussian output layer. The neural network was trained with learning rate of 5e-4. The learning rate was chosen by conducting grid search over {1e-4, 5e-4, 1e-3}. The reward regressor was trained with early stopping rule where we stop training when the validation error does not decrease for 20 training epochs. And we used the trained weights that showed the lowest validation error during the training. For the train and validation split, 20% of the available offline data was used as the validation data, and the rest of the data was used for the training. For the estimation time, when we test our algorithm on the absolute error domain, our algorithm takes 40 seconds on average to make an OPE estimate with 20k offline data on the i7 CPU with 32GB RAM. The experimental results were made with 100 C2-standard-4 instances on the Google Cloud Platform where each instance has 4 virtual CPUs and 16GB of RAM. The code is available at https://github.com/haanvid/kmis.

**Self-Normalization**    All estimators in the experiments are self-normalized as in the work of Kallus and Zhou [2]. Self-normalization is used because it reduces the estimation variance significantly with the addition of a small bias and yields smaller MSE compared to the estimation without self-normalization [2, 13]. The self-normalized kernel-based IS estimator without a metric is shown in Eq. (41):

$$\hat{\rho}_{\text{norm}}^{K} = \frac{\sum_{i=1}^{N} \frac{r_i}{\pi_b(\mathbf{a}_i \mid \mathbf{s}_i)} K\left(\frac{\mathbf{a}_i - \pi(\mathbf{s}_i)}{h}\right)}{\sum_{i=1}^{N} \frac{1}{\pi_b(\mathbf{a}_i \mid \mathbf{s}_i)} K\left(\frac{\mathbf{a}_i - \pi(\mathbf{s}_i)}{h}\right)}. \tag{41}$$

**No Boundary Bias Correction**    Action space can be bounded, and the kernels used in importance sampling can be extended past the bounds. As the actions of the offline data are only observed inside the bounds, the kernels extending outside the bounds will induce bias. Previous works [2, 5] made a correction for the induced bias by truncating the kernel by the action bounds of each dimension and normalizing the kernel. However, we do not correct the boundary bias for all estimators in this work since we regard the boundary bias as negligible compared to the bias reduced by our locally learned metric. Removing the boundary bias correction makes kernel-based IS estimators simpler to use than those with corrections, as the implementation of boundary bias correction requires information on the action bounds, and the integration of the kernel within the action bounds.

**Discretized OPE Estimator**    Discretized OPE estimator discretizes the action space bounded by the minimum and maximum value of the behavior actions for each dimension with the given number of bins assigned for each dimension. In our experiment, we discretized each action dimension by 10 intervals resulting in 100 discretized bins in 2D action space. The IS ratio of the discretized OPE estimator is composed of the indicator function in the numerator and the behavior policy probability density function integrated over the bin in the denominator.

## E.1    Synthetic Domains Experiments

For synthetic domains, we used dropout after each hidden layer with dropout rate of 0.5. Behavior policy density values were clipped below 0.1 for the quadratic reward domain as in the works of Kallus and Zhou [2]. For absolute error domain and multi-modal reward domain, as their behavior policy density is uniform density, we did not apply behavior density value clipping. Clipping the behavior policy density value adds a small bias to the estimation but reduces a significant variance. Therefore, clipping reduces MSEs [2]. For SLOPE, we select a bandwidth from geometrically spaced bandwidths $\{2^{-i} : i \in [1, 7], i \in \mathbb{N}\}$ as in the work of Su et al. [5].

**Quadratic Reward Domain**   The conditional reward distribution given a state $\mathbf{s}$ and an action $\mathbf{a}$ is:

$$p(r\,|\,\mathbf{s},\mathbf{a}) = N\left(r(\mathbf{s},\mathbf{a}), 0.5^2\right), \tag{42}$$

where $r(\mathbf{s},\mathbf{a}) := \mathbb{E}[r\,|\,\mathbf{s},\mathbf{a}]$. The mean reward of a quadratic reward domain given a state $\mathbf{s}$ and an action $\mathbf{a}$ is:

$$r(\mathbf{s},\mathbf{a}) = -(\mathbf{s}-\mathbf{a})^{\top}\left[\begin{array}{cc} 11 & 9 \\ 9 & 11 \end{array}\right](\mathbf{s}-\mathbf{a}). \tag{43}$$

**Absolute Error Domain**   Each dimension of actions and states is sampled uniform randomly in the range of $[-1,1]$. The absolute error which is only dependent on the first dimension of action is designed as:

$$r(\mathbf{s},\mathbf{a}) = -\left|0.5s_1 - a_1\right|, \tag{44}$$

where the reward is not sampled from a distribution but determined by a state $\mathbf{s}$ and an action $\mathbf{a}$.

**Multi-Modal Reward Domain**   Each dimension of actions and states is sampled uniform randomly in the range of $[-1,1]$. The multi-modal reward with exponential functions is designed as:

$$f_1(\mathbf{s},\mathbf{a}) = \exp\left(-\left(\left(\frac{(s_1-a_1)-0.5}{0.25}\right)^2 + \left(\frac{(s_2-a_2)}{1}\right)^2\right)\right),$$

$$f_2(\mathbf{s},\mathbf{a}) = \exp\left(-\left(\left(\frac{(s_1-a_1)+0.5}{0.25}\right)^2 + \left(\frac{(s_2-a_2)}{1}\right)^2\right)\right),$$

$$f_3(\mathbf{s},\mathbf{a}) = \exp\left(-\left(\left(\frac{(s_1-a_1)}{1}\right)^2 + \left(\frac{(s_2-a_2)+0.5}{0.25}\right)^2\right)\right),$$

$$f_4(\mathbf{s},\mathbf{a}) = \exp\left(-\left(\left(\frac{(s_1-a_1)}{1}\right)^2 + \left(\frac{(s_2-a_2)-0.5}{0.25}\right)^2\right)\right),$$

$$r(\mathbf{s},\mathbf{a}) = -\max\left(f_1(\mathbf{s},\mathbf{a}), f_2(\mathbf{s},\mathbf{a}), f_3(\mathbf{s},\mathbf{a}), f_4(\mathbf{s},\mathbf{a})\right), \tag{45}$$

where the reward is not sampled from a distribution but determined by a state $\mathbf{s}$ and an action $\mathbf{a}$.

### E.2   Warfarin Data Experiments

For the Warfarin data, we used L2 regularizer for the 2 hidden layers with the coefficient of 1e-1. The L2 regularizer coefficient was chosen by conducting grid search over $\{$1e-5, 1e-4, 1e-3, 1e-2, 1e-1$\}$. We selected the L2 regularization coefficient that has a lowest validation error (the offline dataset $D$ was splitted into train and validation datasets) for the reward regressor. Behavior policy density values less than 1e-1 were clipped as in the work of Kallus and Zhou [2]. For SLOPE, we selected a bandwidth from $\left\{2^{-i} : i \in [-2,7], i \in \mathbb{N}\right\}$ similar to the work of Su et al. [5].

# F Theoretical Analysis

## F.1 Effect of the Hessian Estimation Error on the Estimation Error and the Convergence Speed of the KMIS Metric Applied Kernel-Based IS Estimator

**Error Analysis**    As $C_v$ of the leading-order variance in Eq. (2) does not change by applying a metric to a kernel due to the constraint $|A(\mathbf{s})| = 1$, the LOMSE of a kernel-based IS OPE estimator with bandwidth $h$ and metric $A$ can be derived as in Eq. (46) by replacing $C_b$ in Eq. (2) with $C_{b,A}$ in Eq. (6)).

$$\text{LOMSE}(h, A, N, D_A) = \underbrace{h^4 C_{b,A}}_{=:\text{LOBIAS}(h,A)^2} + \frac{C_v}{Nh^{D_A}}, \tag{46}$$

$$C_{b,A} := \frac{1}{4} \mathbb{E}_{\mathbf{s} \sim p(\mathbf{s})} \left[ \text{tr} \left( A(\mathbf{s})^{-1} H(\mathbf{s}) \right) \right]^2,$$

$$C_v := R(K) \mathbb{E}_{\mathbf{s} \sim p(\mathbf{s})} \left[ \frac{\mathbb{E}[r^2 \,|\, \mathbf{s}, \mathbf{a} = \pi(\mathbf{s})]}{\pi_b(\mathbf{a} = \pi(\mathbf{s}) \,|\, \mathbf{s})} \right].$$

To reduce the LOMSE, our proposed method learns the metric that minimizes $U_{b,A}$ (Eq. (7)), which is the upper bound of $C_{b,A}$. For the minimization of $U_{b,A}$, we derived the optimal metric matrix $A^*(\mathbf{s})$ given the true Hessian $H(\mathbf{s}) := \mathbf{H_a} \mathbb{E}[r \,|\, \mathbf{s}, \mathbf{a}]|_{a=\pi(s)}$ in Appendix B.2. However, since we will be using the estimated Hessian matrix $\widetilde{H}(\mathbf{s})$ acquired from a reward regressor to compute the estimated optimal metric matrix $\widetilde{A}(\mathbf{s})$, we analyze how the error in $\widetilde{H}(\mathbf{s})$ affects the LOMSE of the KMIS metric applied IS estimation.

In Appendix B.2, we proved that when $H(\mathbf{s})$ contains both positive and negative eigenvalues, the optimal metric makes the squared leading-order bias $\text{LOBIAS}(h, A^*)^2$ zero by making the the trace term $\text{tr}(A^*(\mathbf{s})^{-1} H(\mathbf{s}))$ zero. In this section, we analyze the upper bound of $\text{LOBIAS}(h, \widetilde{A})^2$ in a similar setting where the $H(\mathbf{s})$ has both positive and negative eigenvalues without zero eigenvalues. For our analysis, we use the following notations regarding the true Hessian $H(\mathbf{s})$, the optimal metric matrix computed from the true Hessian $A^*(\mathbf{s})$, estimated Hessian from the reward regressor $\widetilde{H}(\mathbf{s})$, and the optimal metric matrix computed from the estimated Hessian $\widetilde{A}(\mathbf{s})$:

$$H(\mathbf{s}) = [U_+(\mathbf{s}) U_-(\mathbf{s})] \begin{pmatrix} \Lambda_+(\mathbf{s}) & 0 \\ 0 & \Lambda_-(\mathbf{s}) \end{pmatrix} [U_+(\mathbf{s}) U_-(\mathbf{s})]^\top, \tag{47}$$

$$A^*(\mathbf{s}) = \alpha(\mathbf{s}) [U_+(\mathbf{s}) U_-(\mathbf{s})] \begin{pmatrix} d_+(\mathbf{s}) \Lambda_+(\mathbf{s}) & 0 \\ 0 & -d_-(\mathbf{s}) \Lambda_-(\mathbf{s}) \end{pmatrix} [U_+(\mathbf{s}) U_-(\mathbf{s})]^\top, \tag{48}$$

$$\Lambda(\mathbf{s}) = \begin{pmatrix} \Lambda_+(\mathbf{s}) & 0 \\ 0 & \Lambda_-(\mathbf{s}) \end{pmatrix},$$

$$\Lambda_+(\mathbf{s}) = \begin{pmatrix} \lambda_{+,1}(\mathbf{s}) & 0 & \cdots \\ 0 & \lambda_{+,2}(\mathbf{s}) & \\ \vdots & & \ddots \end{pmatrix},$$

$$U_+(\mathbf{s}) = \begin{pmatrix} | & | & \\ \mathbf{u}_{+,1}(\mathbf{s}) & \mathbf{u}_{+,2}(\mathbf{s}) & \cdots \\ | & | & \end{pmatrix},$$

$$\alpha(\mathbf{s}) = \left\{ \left( d_+(\mathbf{s})^{d_+(\mathbf{s})} \prod_{i=1}^{d_+(\mathbf{s})} \lambda_{+,i}(\mathbf{s}) \right) \left( (-1)^{d_-(\mathbf{s})} d_-(\mathbf{s})^{d_-(\mathbf{s})} \prod_{i=1}^{d_-(\mathbf{s})} \lambda_{-,i}(\mathbf{s}) \right) \right\}^{-\frac{1}{d_+(\mathbf{s}) + d_-(\mathbf{s})}}, \tag{49}$$

$$\widetilde{H}(\mathbf{s}) = \left[\widetilde{U}_+(\mathbf{s})\widetilde{U}_-(\mathbf{s})\right] \begin{pmatrix} \widetilde{\Lambda}_+(\mathbf{s}) & 0 \\ 0 & \widetilde{\Lambda}_-(\mathbf{s}) \end{pmatrix} \left[\widetilde{U}_+(\mathbf{s})\widetilde{U}_-(\mathbf{s})\right]^\top, \tag{50}$$

$$\widetilde{A}(\mathbf{s}) = \widetilde{\alpha}(\mathbf{s}) \left[\widetilde{U}_+(\mathbf{s})\widetilde{U}_-(\mathbf{s})\right] \begin{pmatrix} d_+(\mathbf{s})\widetilde{\Lambda}_+(\mathbf{s}) & 0 \\ 0 & -d_-(\mathbf{s})\widetilde{\Lambda}_-(\mathbf{s}) \end{pmatrix} \left[\widetilde{U}_+(\mathbf{s})\widetilde{U}_-(\mathbf{s})\right]^\top, \tag{51}$$

$$\widetilde{\Lambda}(\mathbf{s}) = \begin{pmatrix} \widetilde{\Lambda}_+(\mathbf{s}) & 0 \\ 0 & \widetilde{\Lambda}_-(\mathbf{s}) \end{pmatrix},$$

$$\widetilde{\alpha}(\mathbf{s}) = \left\{ \left( d_+(\mathbf{s})^{d_+(\mathbf{s})} \prod_{i=1}^{d_+(\mathbf{s})} \widetilde{\lambda}_{+,i}(\mathbf{s}) \right) \left( (-1)^{d_-(\mathbf{s})} d_-(\mathbf{s})^{d_-(\mathbf{s})} \prod_{i=1}^{d_-(\mathbf{s})} \widetilde{\lambda}_{-,i}(\mathbf{s}) \right) \right\}^{-\frac{1}{d_+(\mathbf{s})+d_-(\mathbf{s})}} \tag{52}$$

$$= \left( (-1)^{d_-(\mathbf{s})} d_+(\mathbf{s})^{d_+(\mathbf{s})} d_-(\mathbf{s})^{d_-(\mathbf{s})} |\widetilde{H}(\mathbf{s})| \right)^{-\frac{1}{D_A}}$$

$$> 0,$$

where we used eigendecomposition on $H(\mathbf{s})$. $\Lambda_+(\mathbf{s})$ and $\Lambda_-(\mathbf{s})$ are diagonal matrices containing positive and negative eigenvalues, respectively. $U_+(\mathbf{s})$ and $U_-(\mathbf{s})$ are matrices containing eigenvectors corresponding to $\Lambda_+(\mathbf{s})$ and $\Lambda_-(\mathbf{s})$, respectively. Similarly, $\widetilde{\Lambda}_+(\mathbf{s})$, $\widetilde{\Lambda}_-(\mathbf{s})$, $\widetilde{U}_+(\mathbf{s})$, $\widetilde{U}_-(\mathbf{s})$ are decomposed from $\widetilde{H}(\mathbf{s})$. We assume that $\widetilde{H}(\mathbf{s})$ correctly estimates the sign (positive, negative signs) of the eigenvalues in $H(\mathbf{s})$. Because of the assumption, there are $d_+(\mathbf{s})$ and $d_-(\mathbf{s})$ in Eq. (51) and Eq. (52) instead of $\tilde{d}_+(\mathbf{s})$ and $\tilde{d}_-(\mathbf{s})$. We quantify the error in $\widetilde{H}(\mathbf{s})$ with constants $\epsilon$ and $\eta$ ($\epsilon \geq 0$, $\eta > 0$) by:

$$|\widetilde{\mathbf{u}}_{a,i}(\mathbf{s})^\top \mathbf{u}_{a,i}(\mathbf{s}) - 1| \leq \epsilon, \tag{53}$$

$$|\widetilde{\mathbf{u}}_{a,i}(\mathbf{s})^\top \mathbf{u}_{b,j}(\mathbf{s})| \leq \epsilon, \text{ if } a \neq b, \text{ or } i \neq j, \tag{54}$$

$$\|\widetilde{\Lambda}(\mathbf{s})^{-1}\Lambda(\mathbf{s}) - I\| \leq \epsilon, \tag{55}$$

$$0 < \frac{\lambda_{a,i}(\mathbf{s})}{\widetilde{\lambda}_{a,j}(\mathbf{s})} \leq \eta, \text{ if } i \neq j, \tag{56}$$

$$-\eta \leq \frac{\lambda_{a,i}(\mathbf{s})}{\widetilde{\lambda}_{b,j}(\mathbf{s})} < 0, \text{ if } a \neq b, \tag{57}$$

$$\text{where } a, b \in \{+, -\},$$
$$i, j \in \{1, 2, 3, \cdots\}.$$

The trace term $\mathrm{tr}(\widetilde{A}(\mathbf{s})^{-1}H(\mathbf{s}))$ in the LOBIAS$(h, \widetilde{A})^2$ is,

$$\mathrm{tr}(\widetilde{A}(\mathbf{s})^{-1}H(\mathbf{s}))$$

$$= \frac{1}{\widetilde{\alpha}(\mathbf{s})} \mathrm{tr}\left( [d_+(\mathbf{s})\widetilde{U}_+\widetilde{\Lambda}_+\widetilde{U}_+^\top - d_-\widetilde{U}_-\widetilde{\Lambda}_-\widetilde{U}_-^\top]^{-1}[U_+\Lambda_+U_+^\top + U_-\Lambda_-U_-^\top] \right) \tag{58}$$

$$= \frac{1}{\widetilde{\alpha}(\mathbf{s})} \mathrm{tr}\left( \frac{1}{d_+}\widetilde{U}_+\widetilde{\Lambda}_+^{-1}\widetilde{U}_+^\top U_+\Lambda_+U_+^\top + \frac{1}{d_+}\widetilde{U}_+\widetilde{\Lambda}_+^{-1}\widetilde{U}_+^\top U_-\Lambda_-U_-^\top \right. \tag{59}$$

$$\left. - \frac{1}{d_-}\widetilde{U}_-\widetilde{\Lambda}_-^{-1}\widetilde{U}_-^\top U_+\Lambda_+U_+^\top - \frac{1}{d_-}\widetilde{U}_-\widetilde{\Lambda}_-^{-1}\widetilde{U}_-^\top U_-\Lambda_-U_-^\top \right)$$

$$= \underbrace{\frac{1}{\widetilde{\alpha}(\mathbf{s})d_+(\mathbf{s})} \left[ \sum_{j=1}^{d_+(\mathbf{s})}\sum_{i=1}^{d_+(\mathbf{s})} \frac{\lambda_{+,i}(\mathbf{s})}{\widetilde{\lambda}_{+,j}(\mathbf{s})}(\widetilde{\mathbf{u}}_{+,j}(\mathbf{s})^\top \mathbf{u}_{+,i}(\mathbf{s}))^2 + \sum_{j=1}^{d_+(\mathbf{s})}\sum_{i=1}^{d_-(\mathbf{s})} \frac{\lambda_{-,i}(\mathbf{s})}{\widetilde{\lambda}_{+,j}(\mathbf{s})}(\widetilde{\mathbf{u}}_{+,j}(\mathbf{s})^\top \mathbf{u}_{-,i}(\mathbf{s}))^2 \right]}_{=:X(\mathbf{s})}$$

$$\underbrace{- \frac{1}{\widetilde{\alpha}(\mathbf{s})d_-(\mathbf{s})} \left[ \sum_{j=1}^{d_-(\mathbf{s})}\sum_{i=1}^{d_+(\mathbf{s})} \frac{\lambda_{+,i}(\mathbf{s})}{\widetilde{\lambda}_{-,j}(\mathbf{s})}(\widetilde{\mathbf{u}}_{-,j}(\mathbf{s})^\top \mathbf{u}_{+,i}(\mathbf{s}))^2 + \sum_{j=1}^{d_-(\mathbf{s})}\sum_{i=1}^{d_-(\mathbf{s})} \frac{\lambda_{-,i}(\mathbf{s})}{\widetilde{\lambda}_{-,j}(\mathbf{s})}(\widetilde{\mathbf{u}}_{-,j}(\mathbf{s})^\top \mathbf{u}_{-,i}(\mathbf{s}))^2 \right]}_{=:Y(\mathbf{s})}, \tag{60}$$

where in the third equality, we used:

$$\mathrm{tr}(\widetilde{U}_a \widetilde{\Lambda}_a^{-1} \widetilde{U}_a^\top U_b \Lambda_b U_b^\top) = \sum_{j=1}^{d_a(\mathbf{s})} \sum_{i=1}^{d_b(\mathbf{s})} \frac{\lambda_{b,i}(\mathbf{s})}{\widetilde{\lambda}_{a,j}(\mathbf{s})} (\widetilde{\mathbf{u}}_{a,j}(\mathbf{s})^\top \mathbf{u}_{b,i}(\mathbf{s}))^2, \text{ where } a, b \in \{+, -\}. \quad (61)$$

By expanding $X(\mathbf{s})$,

$$X(\mathbf{s}) = \frac{1}{\widetilde{\alpha}(\mathbf{s}) d_+(\mathbf{s})} \left[ \sum_{i=1}^{d_+(\mathbf{s})} \frac{\lambda_{+,i}(\mathbf{s})}{\widetilde{\lambda}_{+,i}(\mathbf{s})} (\widetilde{\mathbf{u}}_{+,i}(\mathbf{s})^\top \mathbf{u}_{+,i}(\mathbf{s}))^2 + \sum_{i \neq j}^{(d_+(\mathbf{s}), d_+(\mathbf{s}))} \frac{\lambda_{+,i}(\mathbf{s})}{\widetilde{\lambda}_{+,j}(\mathbf{s})} (\widetilde{\mathbf{u}}_{+,j}(\mathbf{s})^\top \mathbf{u}_{+,i}(\mathbf{s}))^2 \right.$$
$$\left. + \sum_{j=1}^{d_+(\mathbf{s})} \sum_{i=1}^{d_-(\mathbf{s})} \frac{\lambda_{-,i}(\mathbf{s})}{\widetilde{\lambda}_{+,j}(\mathbf{s})} (\widetilde{\mathbf{u}}_{+,j}(\mathbf{s})^\top \mathbf{u}_{-,i}(\mathbf{s}))^2 \right]. \quad (62)$$

We get the following inequality for $X(\mathbf{s})$ by applying Eq. (53-57):

$$\underbrace{\frac{1}{\widetilde{\alpha}(\mathbf{s})} \left\{ (1-\epsilon)^3 - d_-(\mathbf{s})\eta\epsilon^2 \right\}}_{=:X_l(\mathbf{s})} \leq X(\mathbf{s}) \leq \underbrace{\frac{1}{\widetilde{\alpha}(\mathbf{s})} \left\{ (1+\epsilon)^3 + (d_+(\mathbf{s})-1)\eta\epsilon^2 \right\}}_{=:X_u(\mathbf{s})}. \quad (63)$$

Similarly, we get,

$$\underbrace{\frac{1}{\widetilde{\alpha}(\mathbf{s})} \left\{ (1-\epsilon)^3 - d_+(\mathbf{s})\eta\epsilon^2 \right\}}_{=:Y_l(\mathbf{s})} \leq Y(\mathbf{s}) \leq \underbrace{\frac{1}{\widetilde{\alpha}(\mathbf{s})} \left\{ (1+\epsilon)^3 + (d_-(\mathbf{s})-1)\eta\epsilon^2 \right\}}_{=:Y_u(\mathbf{s})}. \quad (64)$$

The trace term $\mathrm{tr}(\widetilde{A}(\mathbf{s})^{-1} H(\mathbf{s})) (= X(\mathbf{s}) - Y(\mathbf{s}))$ satisfies the following inequality:

$$X_l(\mathbf{s}) - Y_u(\mathbf{s}) \leq \mathrm{tr}(\widetilde{A}(\mathbf{s})^{-1} H(\mathbf{s})) \leq X_u(\mathbf{s}) - Y_l(\mathbf{s}). \quad (65)$$

Then the absolute value of the trace term satisfies the following inequality:

$$|\mathrm{tr}(\widetilde{A}(\mathbf{s})^{-1} H(\mathbf{s}))| \leq \max\{|X_u(\mathbf{s}) - Y_l(\mathbf{s})|, |X_l(\mathbf{s}) - Y_u(\mathbf{s})|\}, \quad (66)$$

In Eq. (66), $|X_u(\mathbf{s}) - Y_l(\mathbf{s})|, |X_l(\mathbf{s}) - Y_u(\mathbf{s})|$ are,

$$|X_u(\mathbf{s}) - Y_l(\mathbf{s})| = \frac{1}{\widetilde{\alpha}(\mathbf{s})} \left\{ 6\epsilon + 2\epsilon^3 + \eta\epsilon^2(2d_+(\mathbf{s}) - 1) \right\}, \quad (67)$$

$$|X_l(\mathbf{s}) - Y_u(\mathbf{s})| = \frac{1}{\widetilde{\alpha}(\mathbf{s})} \left\{ 6\epsilon + 2\epsilon^3 + \eta\epsilon^2(2d_-(\mathbf{s}) - 1) \right\}, \quad \text{where } d_+(\mathbf{s}), d_-(\mathbf{s}) \geq 1, \epsilon \geq 0, \eta > 0. \quad (68)$$

By defining $d_{max}(\mathbf{s}) := \max\{d_+(\mathbf{s}), d_-(\mathbf{s})\}$, we can rewrite Eq. (66):

$$|\mathrm{tr}(\widetilde{A}(\mathbf{s})^{-1} H(\mathbf{s}))| \leq \frac{1}{\widetilde{\alpha}(\mathbf{s})} \left\{ \eta\epsilon^2(2d_{max}(\mathbf{s}) - 1) + 6\epsilon + 2\epsilon^3 \right\}. \quad (69)$$

Then we get the following upper bound of $\mathrm{LOBIAS}(h, \widetilde{A}(\mathbf{s}))^2$:

$$\mathrm{LOBIAS}(h, \widetilde{A})^2 \leq \frac{h^4}{4\widetilde{\alpha}(\mathbf{s})^2} \left[ \eta\epsilon^2(2d_{max}(\mathbf{s}) - 1) + 6\epsilon + 2\epsilon^3 \right]^2. \quad (70)$$

When there is no error in the Hessian estimation acquired from the reward regressor (when $\epsilon = 0$), the upper bound of $\mathrm{LOBIAS}(h, \widetilde{A})^2$ in Eq. (70) is zero. Thus, $\mathrm{LOBIAS}(h, \widetilde{A})^2$ is zero, and this is consistent with what we have derived in Appendix B.2 with $H(\mathbf{s})$. But when there is an error in the Hessian estimation (when $\epsilon > 0$), the $\mathrm{LOBIAS}(h, \widetilde{A})^2$ of the KMIS estimator is upper bounded by $\frac{h^4}{4\widetilde{\alpha}(\mathbf{s})^2} \left[ \eta\epsilon^2(2d_{max}(\mathbf{s}) - 1) + 6\epsilon + 2\epsilon^3 \right]^2$. Therefore, the $\mathrm{LOMSE}(h, \widetilde{A}, N, D_A)$ can be larger than $\mathrm{LOMSE}(h, A^*, N, D_A)$ up to $\frac{h^4}{4\widetilde{\alpha}(\mathbf{s})^2} \left[ \eta\epsilon^2(2d_{max}(\mathbf{s}) - 1) + 6\epsilon + 2\epsilon^3 \right]^2$ due to the Hessian estimation error when $H(\mathbf{s})$ has both positive and negative eigenvalues without zero eigenvalues.

**Convergence Speed Analysis** Now we analyze the convergence speed of the KMIS estimator with $\widetilde{H}(\mathbf{s})$ with additional assumptions that $|\widetilde{H}(\mathbf{s})|$, $\eta$, $\epsilon$ are bounded and the optimal bandwidth $h^*$ is given since bandwidth is an input to our algorithm. MSE with bandwidth $h$ and metric $A$ can be derived as in Eq. (71) by replacing $C_b$ in Eq. (31) with $C_{b,A}$ in Eq. (6).

$$\mathrm{MSE}\,(h, A, N, D_A) = \underbrace{h^4 C_{b,A}}_{=:\mathrm{LOBIAS}(h,A)^2} + O\left(h^6\right) + \frac{C_v}{Nh^{D_A}} + O\left(\frac{1}{Nh^{D_A-2}}\right), \tag{71}$$

where $C_v$ does not change by applying a metric to a kernel due to the constraint $|A(\mathbf{s})| = 1$. By using the upper bound of $\mathrm{LOBIAS}(h, \widetilde{A})^2$ in Eq. (70), and also using $h^* = O\left(\left(\frac{D_A}{N}\right)^{\frac{1}{D_A+4}}\right)$ from Eq. (3) on Eq. (71), we get:

$$\mathrm{MSE}(h^*, \widetilde{A}, N, D_A)$$

$$= O\left((h^*)^4\left(\frac{d_{max}(\mathbf{s})}{\widetilde{\alpha}(\mathbf{s})}\right)^2 \eta^2\epsilon^4\right) + O\left((h^*)^6\right) + O\left(\frac{C_v}{N(h^*)^{D_A}}\right) + O\left(\frac{1}{N(h^*)^{D_A-2}}\right) \tag{72}$$

$$= O\left(\left(\frac{D_A}{N}\right)^{\frac{4}{D_A+4}}\left(\frac{d_{max}(\mathbf{s})}{\widetilde{\alpha}(\mathbf{s})}\right)^2 \eta^2\epsilon^4\right) + O\left(\left(\frac{D_A}{N}\right)^{\frac{4}{D_A+4}}\frac{1}{D_A}\right), \text{ where } N \gg D_A. \tag{73}$$

As for the first term of Eq. (73),

$$O\left(\left(\frac{D_A}{N}\right)^{\frac{4}{D_A+4}}\left(\frac{d_{max}(\mathbf{s})}{\widetilde{\alpha}(\mathbf{s})}\right)^2 \eta^2\epsilon^4\right)$$

$$= O\left(\left(\frac{D_A}{N}\right)^{\frac{4}{D_A+4}}(d_{max}(\mathbf{s}))^2 \underbrace{\left(d_+(\mathbf{s})^{d_+(\mathbf{s})}d_-(\mathbf{s})^{d_-(\mathbf{s})}\right)^{\frac{2}{D_A}}}_{=:Q(\mathbf{s})} \eta^2\epsilon^4\right), \tag{74}$$

$$\because |\widetilde{H}(\mathbf{s})| \text{ is bounded.}$$

Assuming that $d_{max}(\mathbf{s}) = d_+(\mathbf{s})$, $Q(\mathbf{s})$ monotonically increases w.r.t. $d_+(\mathbf{s})$.

$$Q(\mathbf{s}) = \left(d_+(\mathbf{s})^{d_+(\mathbf{s})}(D_A - d_+(\mathbf{s}))^{(D_A-d_+(\mathbf{s}))}\right)^{\frac{2}{D_A}},$$

$$\frac{dQ(\mathbf{s})}{d(d_+(\mathbf{s}))} = \frac{2}{D_A}d_+(\mathbf{s})^{\frac{2d_+(\mathbf{s})}{D_A}}(D_A - d_+(\mathbf{s}))^{\frac{2(D_A-d_+(\mathbf{s}))}{D_A}}\left(\ln\frac{d_+(\mathbf{s})}{D_A - d_+(\mathbf{s})}\right)$$

$$\geq 0 \quad \left(\because D_A - 1 \geq d_+(\mathbf{s}) = d_{max}(\mathbf{s}) \geq \frac{D_A}{2}\right). \tag{75}$$

The maximum of $Q(\mathbf{s})$ is (when $d_+(\mathbf{s}) = D_A - 1$)

$$Q(\mathbf{s}) = (D_A - 1)^{2-\frac{2}{D_A}}. \tag{76}$$

The minimum of $Q(\mathbf{s})$ is (when $d_+(\mathbf{s}) = \frac{D_A}{2}$)

$$Q(\mathbf{s}) = \left(\frac{D_A}{2}\right)^2. \tag{77}$$

We can assume $d_{max}(\mathbf{s}) = d_-(\mathbf{s})$ and show that the maximum and minimum of $Q(\mathbf{s})$ are the same. Therefore,

$$Q(\mathbf{s}) = O\left((D_A)^2\right). \tag{78}$$

With Eq. (78) and using the relation $D_A - 1 \geq d_{max}(\mathbf{s}) \geq \frac{D_A}{2}$ on Eq. (74),

$$\mathcal{O}\left(\left(\frac{D_A}{N}\right)^{\frac{4}{D_A+4}} \left(\frac{d_{max}(\mathbf{s})}{\widetilde{\alpha}(\mathbf{s})}\right)^2 \eta^2 \epsilon^4\right) = \mathcal{O}\left(\left(\frac{D_A}{N}\right)^{\frac{4}{D_A+4}} (D_A)^4 \eta^2 \epsilon^4\right). \tag{79}$$

By plugging in Eq. (79) to Eq. (73),

$$\mathrm{MSE}(h^*, \widetilde{A}, N, D_A) = \mathcal{O}\left(\left(\frac{D_A}{N}\right)^{\frac{4}{D_A+4}} (D_A)^4 \eta^2 \epsilon^4\right) + \mathcal{O}\left(\left(\frac{D_A}{N}\right)^{\frac{4}{D_A+4}} \frac{1}{D_A}\right). \tag{80}$$

When the estimated Hessian $\widetilde{H}(\mathbf{s})$ has no error (when $\epsilon = 0$), the convergence speed derived in Eq. (80) matches the convergence speed of the KMIS estimator with a true Hessian derived in Appendix B.3.

However, when there is an error in the estimated Hessian (when $\epsilon > 0$), MSE of a KMIS estimator with the estimated Hessian converges to zero slower by rate of $\mathcal{O}((D_A)^5)$ compared to the KMIS with a true Hessian:

$$\frac{\mathrm{MSE}(h^*, \widetilde{A}, N, D_A)}{\mathrm{MSE}(h^*, A^*, N, D_A)} = \mathcal{O}\left((D_A)^5\right). \tag{81}$$

Therefore, the KMIS estimator with the estimated Hessian has a slower convergence rate to a true policy value than the KMIS estimator with a true Hessian by the rate of $\mathcal{O}((D_A)^{\frac{5}{2}})$.