# OpenReview forum: "Local Metric Learning for Off-Policy Evaluation in Contextual Bandits with Continuous Actions"
_NeurIPS.cc/2022/Conference — NeurIPS 2022 Accept_

### Official Review · Reviewer_PWoc · 2022-06-22

**Rating:** 6
**Confidence:** 4
**Soundness:** 3 good
**Presentation:** 2 fair
**Contribution:** 3 good

**Summary:**

The paper studies the problem of off-policy evaluation of deterministic target policies for continuous actions. In this setting, the typical importance sampling approach does not work, so previous papers deal with the problem by relaxing the target policy via the kernel smoothing technique. The authors improve the existing kernel smoothing, especially for multi-dimensional continuous action setting, by proposing a method to data-drivenly learn a Mahalanobis distance metric in the action space. More specifically, the metric is learned to minimize the bias of the resulting estimator. The authors provide some theoretical analysis of the proposed MIS estimator showing that MIS converges faster than IS without metric learning. Moreover, some empirical analysis is performed demonstrating that MIS is more accurate in terms of MSE for a range of bandwidths compared to the baseline estimators.

**Questions:**

- Could you compare the convergence rate of the proposed method and the estimator based on the discretization proposed by [Cai et al. 2021]? [Cai et al. 2021] compares their discretization approach with the kernel smoothing approach in their theoretical analysis.

- Did you perform an experiment to compare the estimators with increasing dimensions of the action space? There should be this kind of experiment given the discussion around Proposition 3.3 in Section 3.

- Did you use the same reward regressor for DM and MIS? I was wondering where the large difference between these two estimators comes from, even though both estimators rely on the estimated expected reward. Some explanation would be appreciated.

- Related to the above question, how does MIS work when you gradually increase the noise on the reward? (the reward function estimation will become gradually difficult)

- Does the proposed method improve the bias or the variance in the experiments? I was curious about the bias-variance decomposition in the experiments. So, it would be great to have additional figures about the bias-variance decomposition, at least in the appendix.

- Why does DM degrade its accuracy with increasing sample size in Figure 3 (a)?

[Cai et al. 2021] Hengrui Cai, Chengchun Shi, Rui Song, and Wenbin Lu. Deep jump learning for off-policy evaluation in continuous treatment settings. Advances in Neural Information Processing Systems, 34, 2021.


**Limitations:**

The proposed approach is not compared with the discretization approach of [Cai et al. 2021], neither theoretically nor empirically.

There is also no related work section, even though there seems to be some space enough to include one additional section. It would be great to have some discussion on the comparison between the kernel smoothing method (authors’ approach) and discretization for continuous action OPE. Moreover, it would be useful to summarize a broader literature of OPE for contextual bandits and reinforcement learning.

An acronym “MIS” has already been used for “Marginalized Importance Sampling”, which is an OPE estimator for reinforcement learning proposed by [Xie et al. 2019].

- [Xie et al. 2019] Tengyang Xie, Yifei Ma, Yu-Xiang Wang. Towards Optimal Off-Policy Evaluation for Reinforcement Learning with Marginalized Importance Sampling. Advances in Neural Information Processing Systems, 32, 2019.

Typos
- L46, a bug in citation
- Eq 5, an unnecessary bracket in the LHS of the equation
- L144, presented in 4 -> presented in Eq.(4) ?
- L246, $a_{min}, a_{max}$


=== after the response ===

The author response, additional experiments, and the new acronym of the proposed method seem reasonable to me, so I increase my score to 6. Please definitely add the related work section, in particular, a comparison with [Cai et al. 2021] would be helpful to the readers.

**Strengths And Weaknesses:**

Strengths
- makes a solid improvement of the kernel smoothing technique used for continuous action OPE by learning a distance metric from the data
- theoretical and empirical evidence are provided to support the superior OPE performance of the proposed approach compared to the conventional approach without any distance metric

Weaknesses
- the proposed estimator needs an accurate estimation of the expected reward function, and there seems to be no theoretical analysis on how the estimation error of the expected reward function affects the OPE accuracy of the resulting estimator
- there is no theoretical and empirical comparison between the proposed method and the discretization approach of [Cai et al. 2021]
- Some important experiment configurations are missing (see below)

---

> ### Author Response · Authors · 2022-08-04
> **Response to Reviewer PWoc [1]**
>
>
>
> **[Experiment on increasing noise of the reward]**
>
> We added the experimental result in Appendix A.4 Figure 5. showing the performances of DM and our algorithm when the noise in the rewards increases.
>
> **[Performance degradation of DM in Figure 3 (a) when the sample size increases]**
>
> The reviewer asked why the performance of DM degrades as data size increases in Figure 3 (a). Similar results can be observed in Figure 1. of Kallus and Zhou’s paper [Kallus and Zhou. 2018]. The performance of the polynomial regression reported in the figure is degraded while the data size is increased. This may be due to the covariate shift problem often observed in OPE problem settings. Train data and test data in OPE problem settings have different distributions since they are generated by different policies (behavior policy and target policy). This causes the DM to get biased, and as pointed out in the works of Jiang et al. [Jiang et al. 2016], such bias is hard to analyze.
>
>
> **[Reward regressor used in DM and MIS]**
>
> The reviewer asked whether we use the same reward regressor for both DM and our algorithm. The difference between DM and the proposed algorithm is that DM directly estimates the expected reward given a state and action with a reward regressor. For our algorithm, it acquires only the Hessian information of the reward from the reward regressor and uses it to compute Mahalanobis distance metrics. Observing the empirical results in Figure 2 and 3, we conjecture that learning the Hessian information is more accessible than learning the whole reward function since the MSE of DM is larger than KMIS applied methods.
>
>
>
> **[Typos]**
>
> We fixed the bug in the citation and other typos the reviewer pointed out and colored the changed texts in red.
>
> [Kallus and Zhou. 2018] Kallus, Nathan, and Angela Zhou. "Policy evaluation and optimization with continuous treatments." AISTATS. 2018.
>
> [Jiang et al. 2016] Jiang, Nan, and Lihong Li. "Doubly robust off-policy value evaluation for reinforcement learning." ICML, 2016.

---

> ### Author Response · Authors · 2022-08-04
> **Response to Reviewer PWoc  [2]**
>
>
> **[Comparison between our method and  deep jump learning (DJL) [Cai et al. 2021] ]**
>
> The reviewer asked for a comparison between our proposed model and deep jump learning (DJL) [Cai et al. 2021]. DJL and our proposed model cannot be compared directly since DJL focuses on environments with a single action dimension (mentioned in Section 2.1 in their paper) while our algorithm focuses on environments with two or more action dimensions. Metric learning requires two or more action dimensions, and thus, we cannot use our algorithm on the problem setting of DJL. One may alternatively extend DJL’s environments to larger-than-one action dimensions, but it is also not straightforward since DJL’s discretization method is specialized to one-dim action spaces. Also, the authors of DJL compared across kernel smoothing methods, but they only considered environments with a single continuous action dimension where our algorithm is not applicable.
>
>
> **[Show bias-variance decomposition in the experiments ]**
>
> To address the request of the reviewer regarding an experiment showing bias-variance decomposition, we conducted an experiment showing bias-variance decomposition in Figure 3.
>
>
> **[Addition of a related works section ]**
>
> The reviewer asked for more literature review. Due to the page limit, we could not add the related works section. But we will add in the related works section later in the camera ready if we get accepted.
>
> To briefly mention the recent related works on OPE for contextual bandits with continuous actions here, there are a group of kernel-based methods [Su et al. 2020][Kallus and Zhou. 2018] and a discretization method [Cai et al. 2021]. For the kernel-based methods, Su et al. proposed the algorithm “Selection by Lepski’s principle for Off-Policy Evaluation” (SLOPE) that could be used for a bandwidth selection of a kernel-based OPE estimator [Cai et al. 2021]. The algorithm is based on Lepski’s principle for bandwidth selection in nonparametric statistics [Lepski. 1992]. And Kallus and Zhou proposed learning a bandwidth that minimizes the leading order MSE of a kernel-based OPE estimation [Kallus and Zhou. 2018]. We have compared these recent kernel-based algorithms in our work. For the discretization method, Cai et al. proposed an algorithm that adaptively discretizes a one-dimensional continuous action space. The action space is discretized to have similar expected rewards for each discretized interval given a state. They point out that the kernel-based methods use a single bandwidth, even though MSE minimizing bandwidth may vary across the action space. And show that their adaptive discretization method performs better in such cases. It would be interesting to compare their algorithm to ours since our method learns a kernel metric that controls the "relative" bandwidth of a kernel at each state along the direction of eigenvectors of the metric matrix rather than just using an isotropic metric. However, their work focuses on domains with a single action dimension, and extending their work to our problem setting with two or more action dimensions is not straightforward. Therefore their work is not directly comparable to ours.
>
> **[Issue on the acronym of our algorithm]**
>
> To address the issue of using the same acronym with the “Marginalized Importance Sampling” [Xie et al. 2019] raised by the reviewer, We changed from using “MIS” to “KMIS” to avoid the issue.
>
> **[Experiment on increasing action dimension]**
>
> We added experimental results on increasing action dimension in the absolute error domain in Figure 3. We plot the empirical biases and MSEs of a kernel relaxed IS estimator with and without our proposed metric to validate the analysis in Proposition 1.
>
>
>
>
> [Cai et al. 2021]  Cai, Hengrui, et al. "Deep jump learning for off-policy evaluation in continuous treatment settings." NeurIPS, 2021.
>
> [Kallus and Zhou. 2018] Kallus, Nathan, and Angela Zhou. "Policy evaluation and optimization with continuous treatments." AISTATS. 2018.
>
> [Su et al. 2020] Su, Yi, Pavithra Srinath, and Akshay Krishnamurthy. "Adaptive estimator selection for off-policy evaluation." ICML, 2020.
>
> [Lepski. 1992] Lepski, O. V. Asymptotically minimax adaptive estimation. i: Upper bounds. optimally adaptive estimates. Theory of Probability & Its Applications, 1992.
>
> [Xie et al. 2019] Tengyang Xie, Yifei Ma, Yu-Xiang Wang. Towards Optimal Off-Policy Evaluation for Reinforcement Learning with Marginalized Importance Sampling. NeurIPS 2019.

---

> ### Author Response · Authors · 2022-08-05
> **Response to Reviewer PWoc [3]**
>
> Thank you for your detailed feedback.
>
> **[MSE and convergence speed affected by the error in the reward regressor]**
>
> We added an analytic analysis on how the error in the reward regressor (which is used for a Hessian estimation in our algorithm) affects the estimation error and the convergence speed of our proposed method in Appendix A.10

---

### Official Review · Reviewer_9JXx · 2022-06-30

**Rating:** 7
**Confidence:** 2
**Soundness:** 3 good
**Presentation:** 4 excellent
**Contribution:** 4 excellent

**Summary:**

This paper focuses on off-policy evaluation in contextual bandits with continuous action space. To measure the similarity of a behavior action to a target action according to the difference of rewards, the paper proposes local kernel metric learning for importance sampling (MIS). By analyzing the leading order MSE with optimal bandwidth, the author shows that applying the metric obtained in Theorem 1 to kernel-relaxed IS estimate reduces the MSE. They also show that metric learning increases the convergence speed. Their experimental results verify their statements.

**Questions:**

- By applying the metric learning to kernel-relaxed IS estimate, how much MSE can be reduced with respect to $D$ and the sample sizes $N$? Is there a theoretical guarantee?
- Algorithm 1 applies a neural network reward regressor. How does the error of this regressor affect the MSE and the convergence speed? The author should discuss more on this these and may give some theoretical analysis.
- How to choose $\beta$ and $\gamma$ in equation (11)?


**Limitations:**

Assuming an optimal bandwidth is provided.

**Strengths And Weaknesses:**

Strengths:
- Interesting question. In many real-world problems, we need to measure the similarities of actions according to their resulting rewards.
- Convincing experimental results.
- Well-written and easy to follow. Though I am new in this area, it is not hard for me to get the main idea and follow the discussion in the paper because the paper is well-written.

Weaknesses:
- Lack of theoretical guarantees. See Question below.

---

> ### Author Response · Authors · 2022-08-04
> **Response to Reviewer 9JXx**
>
> Thank you for your thoughtful feedback.
>
> **[Reduced leading order MSE due to metric learning w.r.t. N and D]**
>
> The reviewer asked how much MSE is reduced with our proposed metric w.r.t. the sample size $N$ and the action dimension $D$. Assuming that the hessian of an expected reward w.r.t. an action contains both positive and negative eigenvalues, the bias term in the LOMSE (first term in Eq.(3)) becomes zero and only the leading order variance (second term in Eq.(3)) remains which are represented w.r.t. $N$ and $D$. Notice that our metric matrix does not affect the leading order variance in Eq.(3) when action $\mathbf{a}$ in Eq.(3) is transformed to $\mathbf{z}=L(\mathbf{s})^T \mathbf{a}$ by our metric matrix $A(\mathbf{s}) (=L(\mathbf{s})L(\mathbf{s})^T)$ as mentioned in Section 3.
>
> We have revised the paper in Section 3, line 159~161 in page 5 to explain the effect of our metric on the leading order MSE w.r.t. $N$ and $D$.
>
>
> **[How to select $\beta (\mathbf{s})$ and $\gamma(\mathbf{s})$ in Eq.(12)]**
>
> The reviewer asked the details on how to select $\beta (\mathbf{s})$ and $\gamma(\mathbf{s})$ that regularize our derived metric matrix $A(\mathbf{s})$ to have a determinant of 1 and positive definite. To compute the regularizers,
>
> 1. Add a small positive real coefficient $c(\mathbf{s})\mathbb{I}$ to the metric matrix in Eq.(10) to avoid metric matrix $A(\mathbf(s))$ from having a determinant of 0. The coefficient is selected to be the maximum value among eigenvalues in the Hessian of reward multiplied by 0.01. We let the coefficient depend on the state since the scale of eigenvalues of the Hessian matrix differs depending on experiment environments.
> 2. After adding  $c(\mathbf{s})\mathbb{I}$ to the metric matrix, we compute the determinant of the resulting matrix $M(\mathbf{s})$. To scale the metric matrix to have a determinant of 1,
> 3. We scale the matrix by multiplying $\beta(s) = \frac{1}{ |M|^(-1/D) }$    ($D$ : action dimension). Then, $\gamma(s) = \beta(s)*c(s)$.
>
>
> To address the issue of the missing details raised by the reviewer, we added a description in Appendix A.8.
>
>
>
>
> **[Metric learning and optimal bandwidth]**
>
> The reviewer asked if our proposed algorithm requires an optimal bandwidth as an input. Our algorithm does not require an optimal bandwidth to learn a metric. The metric is learned to minimize the absolute value of the leading order bias in Eq.(9) by minimizing the absolute value of the trace in the equation. Therefore, our algorithm does not require any information on bandwidths.
>
> The reason we introduced optimal bandwidth $h^*$ in Eq.(4) was to explain why it is important to reduce bias in Proposition 1.
>
> Regarding this concern, we have made the following revisions in the paper: line 136 ~ 137, line 146 ~ 149 on pg 4, and line 157~158 on pg 5

---

> ### Author Response · Authors · 2022-08-05
> **Response to Reviewer 9JXx [2]**
>
> **[MSE and convergence speed affected by the error in the reward regressor]**
>
> We added an analytic analysis on how the error in the reward regressor (which is used for a Hessian estimation in our algorithm) affects the estimation error and the convergence speed of our proposed method in Appendix A.10

---

### Official Review · Reviewer_ZjpK · 2022-07-12

**Rating:** 7
**Confidence:** 3
**Soundness:** 3 good
**Presentation:** 3 good
**Contribution:** 3 good

**Summary:**

The paper studies off-policy evaluation in contextual bandits with continuous actions, where prior works estimate a deterministic policy using the estimated performance of a kernelized policy to address the infinity variance problem of inverse propensity weighting in this setting. In particular, they focus on scenarios where the action-space is multi-dimensional, where Euclidean distance might be a poor measure of similarity between actions since the relationship between the reward difference and the distance difference in each direction might be quite different. So they propose to learn a metric of the action space so that similar actions (in terms of rewards) have smaller distance and vice versa.The metric is learned to approximately minimize the mean squared error of the kernelized IPW estimator. They conduct empirical results to show the effectiveness of the proposed method.

**Questions:**

See weaknesses.

**Limitations:**

I did not find limitations nor potential negative societal impact of their work.

**Strengths And Weaknesses:**

1. Selecting a good distance metric for off-policy evaluation in contextual bandits with continuous actions is an interesting problem.

2. The paper is well-written and easy-to-follow.

3. The proposed distance metric learning approach seems to make sense and the paper provides some theoretical analysis on its (approximate) optimality.

Weaknesses

1. Most of the weaknesses I thought of are already mentioned in the conclusion section in the paper. The algorithm seems to require athe optimal bandwidth as input, which is not known; and there are many approximations in the theoretical analysis and the Hessian matrix.

2. I am confused about Algorithm 1. is the kernel and its bandwidth inputs to the algorithm? If yes, please write it explicitly.

---

> ### Author Response · Authors · 2022-08-04
> **Response to Reviewer ZjpK**
>
> We thank you for your positive feedback
>
> **[Metric learning and optimal bandwidth]**
>
> The reviewer asked if our proposed algorithm requires an optimal bandwidth as an input. Our algorithm does not require an optimal bandwidth to learn a metric. The metric is learned to minimize the absolute value of the leading order bias in Eq.(9) by minimizing the absolute value of the trace in the equation. Therefore, our algorithm does not require any information on bandwidths.
>
> The reason we introduced optimal bandwidth $h^*$ in Eq.(4) was to explain why it is important to reduce bias in Proposition 1.
>
> Regarding this concern, we have made the following revisions in the paper: line 136 ~ 137, line 146 ~ 149 on pg 4, and line 157~158 on pg 5
>
> **[Kernel and bandwidth as input to our algorithm]**
>
> The reviewer asked if the bandwidth and kernel are input to our algorithm. Our algorithm takes kernel and bandwidth as inputs. We modified Algorithm 1 to explicitly show that bandwidth and a kernel are inputs to our algorithm in our revision.

---

> > ### Comment · Reviewer_ZjpK · 2022-08-08
> > **Thanks for the clarification**
> >
> > I thank the authors for their response and I would like to keep my evaluation.

---

### Official Review · Reviewer_E6Fs · 2022-07-13

**Rating:** 4
**Confidence:** 4
**Soundness:** 2 fair
**Presentation:** 1 poor
**Contribution:** 2 fair

**Summary:**

Motivated by the fact that target policy should be deterministic in certain real-world scenarios, this paper works on off-policy policy evaluation of deterministic policies in the contextual bandit setting. This work proposes to use local kernel metric learning for importance sampling to derive OPE estimates. For a given state, this proposed metric measures the distance between a behavior policy and a target policy based on the difference of the rewards of them.

**Questions:**

The main idea that motivates the local kernel metric learning for IS is that measuring the distance between actions based on resulting rewards instead of the Euclidean distance. Following this, how could the proposed approach reconcile with the classic attribution issue in reinforcement learning? As we know, the reward following an action may be attributed to more than one previous actions instead of only the most recent action, especially in the sparse reward setting.

**Limitations:**

The authors have mentioned some potential limitations.

**Strengths And Weaknesses:**

This work points out a view that we could measure similarities of different actions based on the rewards generated by them, instead of the Euclidean distances between them. This view motivates the authors to propose to use local kernel metric learning for importance sampling (MIS) for OPE estimates. The overall presentation of the idea and method is not very clear. For example, at the beginning of the introcution section, the authors seems to use offline reinforcement learning (RL) and off-policy RL interchangeably; in line 47, the authors mention the "bias" but fails to define or explain it from lines 41 to 46 or even earlier paragraphs, which is confusing.

While OPE is a critical question in the RL field, there are limited innovations in this paper. Besides, given the recent extensive exploration of OPE for both discrete and continuous action setting, this work misses a significant part of existing work on this topic.

---

> ### Author Response · Authors · 2022-08-04
> **Response to Reviewer E6Fs**
>
> Thank you for your thoughtful feedback
>
> **[Clarity]**
>
> The reviewer asked to clarify whether the paper addresses off-policy or offline. Our approach considers offline setting as in the other off-policy policy evaluation algorithms [Kallus and Zhou. 2018][Su et al. 2020] that we compared in our work. We clarify this in line 23~25 on page 1.
>
> The reviewer also asked for the definition of bias used in the introduction section. We added the definition of bias in Eq.(7) and referred to this definition.
>
> **[Missing review on the related works]**
>
> The reviewer asked for more literature review. Due to the page limit, we could not add the related works section. But we will add in the related works section later in the camera ready if we get accepted.
>
> To briefly mention the recent related works on OPE for contextual bandits with continuous actions here, there are a group of kernel-based methods [Su et al. 2020][Kallus and Zhou. 2018] and a discretization method [Cai et al. 2021]. For the kernel-based methods, Su et al. proposed the algorithm “Selection by Lepski’s principle for Off-Policy Evaluation” (SLOPE) that could be used for a bandwidth selection of a kernel-based OPE estimator [Cai et al. 2021]. The algorithm is based on Lepski’s principle for bandwidth selection in nonparametric statistics [Lepski. 1992]. And Kallus and Zhou proposed learning a bandwidth that minimizes the leading order MSE of a kernel-based OPE estimation [Kallus and Zhou. 2018]. We have compared these recent kernel-based algorithms in our work. For the discretization method, Cai et al. proposed an algorithm that adaptively discretizes a one-dimensional continuous action space. The action space is discretized to have similar expected rewards for each discretized interval given a state. They point out that the kernel-based methods use a single bandwidth, even though MSE minimizing bandwidth may vary across the action space. And show that their adaptive discretization method performs better in such cases. It would be interesting to compare their algorithm to ours since our method learns a kernel metric that controls the "relative" bandwidth of a kernel at each state along the direction of eigenvectors of the metric matrix rather than just using an isotropic metric. However, their work focuses on domains with a single action dimension, and extending their work to our problem setting with two or more action dimensions is not straightforward. Therefore their work is not directly comparable to ours.
>
> **[How to expand our work to policy evaluation in RL cases]**
>
> Expanding our work to an RL case is our future work. A contextual bandit case can be seen as a one-step RL case where a reward is only dependent on one timestep. However, in RL, a reward depends on a history of actions. Therefore, it is not straightforward to expand our work to RL cases. One possible approach for our future work is to use Hessian of an action value w.r.t. an action to learn a metric. We left this as our future work.
>
>
>
> [Cai et al. 2021]  Cai, Hengrui, et al. "Deep jump learning for off-policy evaluation in continuous treatment settings." NeurIPS, 2021.
>
> [Kallus and Zhou. 2018] Kallus, Nathan, and Angela Zhou. "Policy evaluation and optimization with continuous treatments." AISTATS. 2018.
>
> [Su et al. 2020] Su, Yi, Pavithra Srinath, and Akshay Krishnamurthy. "Adaptive estimator selection for off-policy evaluation." ICML, 2020.
>
> [Lepski. 1992] Lepski, O. V. Asymptotically minimax adaptive estimation. i: Upper bounds. optimally adaptive estimates. Theory of Probability & Its Applications, 1992.

---

### Official Review · Reviewer_og43 · 2022-07-14

**Rating:** 7
**Confidence:** 3
**Soundness:** 3 good
**Presentation:** 3 good
**Contribution:** 3 good

**Summary:**

The paper proposes to learn a Mahalanobis distance metric using local information in order to reduce the bias of kernel-based off-policy estimators for deterministic policies in continuous action spaces. The paper proves that when the density is twice differentiable, then the optimal matrix can be computed in closed form from the Hessian, and that when the optimal bandwidth is used their proposed estimator has a faster convergence rate than the alternative without the learned metric. Finally, the paper shows in experiments on synthetic data and the Warfarin Dosing dataset that it consistently improves over baselines.

**Questions:**

1) (minor) Both of the chosen baselines are pretty strong, and the Kallus & Zhou paper already compare to a discretized baseline, but it might be helpful for new audiences to see a comparison with a discretized baseline to establish that both of the chosen baselines beat a simple approach.

**Limitations:**

The paper discusses the limitation that they assume a fixed bandwidth rather than finding a new bandwidth to adapt to the learned metric, and leave it open for future work. The paper also notes that the method is only proven to work for twice-differentiable, but that it empirically seems to work in other cases.

**Strengths And Weaknesses:**

Strengths:
1) The empirical performance is quite strong, which makes it more significant.
2) In terms of significance, the proposed method is fairly general, and can be dropped into any setting where there is a kernel-based approximation to a deterministic policy in a continuous action space.
3) In terms of quality, the paper evaluates the baseline algorithms on the same experiments as were used in the original papers for the baseline, suggesting that the method is able to improve performance on problems where the baselines were already strong.
4) The paper is overall well-written.

Weaknesses:
1) (minor) Clarity could be improved by explaining more of how the proofs work instead of just referring to the appendix.

---

> ### Author Response · Authors · 2022-08-04
> **Response to Reviewer og43**
>
> Thank you for your constructive feedback
>
> **[Clarity]**
> In order to address the clarity issue raised by the reviewer on the proofs of theorems and propositions, we included more details and core ideas on the proofs in Proposition 1, Theorem 1, and Theorem 2.
>
> **[Discretized OPE as a baseline]**
>
> We agree with the reviewer that adding a simple discretized baseline to our experiment as in [Kallus and Zhou, 2018] would help new audiences to see the difference between the baselines and the simple discretization approach. We added the baseline in Figure 2 for comparison.
>
> In Figure 2, the discretized OPE estimator performs better than the other baselines in the absolute error domain and the multi-goal domain where the domains have uniform action distributions and no noise in the rewards. In this condition, because we apply self-normalization to all estimators, the discretized OPE would take an average of the behavior rewards which corresponding behavior actions are in the same discretized bins with the target actions (we used 100 bins resulting from discretizing each action dimension by 10) which can be accurate in this condition.
>
> [Kallus and Zhou, 2018] Kallus, Nathan, and Angela Zhou. "Policy evaluation and optimization with continuous treatments." AISTATS, 2018.

---

### Author Response · Authors · 2022-08-04
**General Response**

Dear Reviewers, thank you very much for your positive feedback on our paper. To summarize the main contribution of our work, we first showed that bias becomes a dominant factor in the MSE of kernel-relaxed importance sampling (IS) off-policy policy evaluation (OPE) estimates. And we derived the kernel metric that minimizes the leading order bias in the MSE.

The improvements and modifications in the revision are summarized below:

**[Analytic Analysis]**

- We added an analytic analysis on how the error in the reward regressor (which is used for a Hessian estimation in our algorithm) affects the estimation error and the convergence speed of our proposed method in Appendix A.10


**[Experiments]**

- (Figure 2)  We added the discretized OPE estimator as a baseline in the synthetic domain experiment.

- (Figure 3) We empirically validate Proposition 1, by experimenting on a synthetic domain with varying action dimensions. As the action dimension increases, the empirical bias of the kernel-relaxed IS OPE estimates becomes a dominant factor in MSE. And we also show that the application of our metric reduces a significant amount of bias.

- (Figure 5 in Appendix A.4) We empirically analyze the performance of our algorithm when the performance of the reward regressor (DM) used for Hessian estimation in our algorithm gradually degrades. This is done by experimenting on a synthetic domain with an increasing level of variance in rewards contained in the data. Due to the noise, the estimation error in DM gradually increases. But our algorithm still shows the best performance among the compared OPE algorithms.

**[Writing]**

- We added in more details and core ideas on the poofs of the theorems and the proposition.
- We revised Section 3 to help readers understand that our algorithm does not require an optimal bandwidth to learn the proposed metric.
- We fixed Algorithm 1 to explicitly show that bandwidth and a kernel are input to our algorithm
- Details on how to select the regularizers $\beta(\mathbf{s})$, $\gamma(\mathbf{s})$ are added in Appendix A.8
- The acronym for our method is changed from "MIS" to "KMIS" to avoid using the same acronym with the “Marginalized Importance Sampling” [Xie et al. 2019]

We have also introduced changes (text colored in red)  in the paper in order to reflect the comments made by reviewers. Individual responses to reviewers are provided below in a separate text box.

[Xie et al. 2019] Tengyang Xie, Yifei Ma, Yu-Xiang Wang. Towards Optimal Off-Policy Evaluation for Reinforcement Learning with Marginalized Importance Sampling. NeurIPS 2019.

---

### Author Response · Authors · 2022-08-09
**General Response [2]**

Dear Reviewers, we derived tighter error bound and corresponding convergence speed of our estimator when there is an error in the reward regressor used for a Hessian estimation in our algorithm and revised Appendix A.10 "Effect of Error in a Reward Regression (Hessian Estimator for KMIS) on the Estimation Error and Convergence Speed of KMIS" in the paper submitted today.

---

### Meta-Review · Area_Chair_Zpk1 · 2022-08-26

**Recommendation:** Accept
**Confidence:** Certain

**Metareview:**

Reviewers and AC are satisfied by the authors' response and think the paper should be accepted.

**Award:**

No

---

### Decision · Program_Chairs · 2022-09-14

Accept